# Regulation of Zbp1 by miR-99b-5p in microglia controls the development of schizophrenia-like symptoms in mice

Lalit Kaurani [1,15 ✉], Md Rezaul Islam[1,15], Urs Heilbronner[2,15], Dennis M Krüger[1], Jiayin Zhou[1], Aditi Methi [1], Judith Strauss[3], Ranjit Pradhan[1], Sophie Schröder[1], Susanne Burkhardt[1], Anna-Lena Schuetz[1], Tonatiuh Pena [1], Lena Erlebach[4], Anika Bühler[4], Monika Budde[2], Fanny Senner[2], Mojtaba Oraki Kohshour[2], Eva C Schulte[2,5,6,7], Max Schmauß[8], Eva Z Reininghaus[9], Georg Juckel[10], Deborah Kronenberg-Versteeg[4], Ivana Delalle[11], Francesca Odoardi[3], Alexander Flügel[3], Thomas G Schulze [2 ✉], Peter Falkai [5 ✉], Farahnaz Sananbenesi [12 ✉] & Andre Fischer [1,13,14 ✉]

## Abstract

**Current approaches to the treatment of schizophrenia have mainly focused on the protein-coding part of the genome; in this context, the roles of microRNAs have received less attention. In the present study, we analyze the microRNAome in the blood and postmortem brains of schizophrenia patients, showing that the expression of miR-99b-5p is downregulated in both the prefrontal cortex and blood of patients. Lowering the amount of miR-99b-5p in mice leads to both schizophrenia-like phenotypes and inflammatory processes that are linked to synaptic pruning in microglia. The microglial miR-99b-5p-supressed inflammatory response requires Z-DNA binding protein 1 (Zbp1), which we identify as a novel miR-99b-5p target. Antisense oligonucleotides against Zbp1 ameliorate the pathological effects of miR-99b-5p inhibition. Our findings indicate that a novel miR-99b-5p-Zbp1 pathway in microglia might contribute to the pathogenesis of schizophrenia.**

**Keywords** Schizophrenia; microRNA; miR-99b; Zbp1; Microglia
**Subject Categories** Immunology; Neuroscience; RNA Biology

## Introduction

Schizophrenia (SZ) is a devastating psychiatric disorder, and the difficulties involved in treating and managing it make it one of the ten most expensive disorders for health care systems worldwide (Kessler et al [2005]). SZ is believed to evolve on the background of complex genome-environment interactions that alter the cellular homeostasis as well as the structural plasticity of brain cells. Thus, genetic predisposition and environmental risk factors seem to affect processes that eventually contribute to the manifestation of clinical symptoms (Nestler et al [2016]; Giegling et al [2017]; Robinson and Bergen [2021]). Despite the available pharmacological and non-pharmacological treatment options, a significant number of patients do not benefit from these treatments in the long-term, underscoring the need for novel and potentially stratified therapeutic approaches (Samara et al, [2016]). So far, drug development has focused on the human transcriptome that encodes proteins, but the success of this approach is limited (Spark et al, [2022]). The transcriptome also consists of non-coding RNAs (ncRNAs) that represent about 98.5% of the entire transcriptome and which are recognized as key regulators of cellular functions (Ezkurdia et al, [2014]). Therefore, RNA therapeutics represent an emerging concept that may expand current therapeutic strategies focused on the protein-coding part of our genome (Damase et al, [2021]; Nemeth et al, [2023]). RNA therapies utilize, for example, antisense oligonucleotides (ASOs), siRNA, microRNA (miR) mimics or corresponding anti-miRs to control the expression of genes and

[1]Department for Epigenetics and Systems Medicine in Neurodegenerative Diseases, German Center for Neurodegenerative Diseases (DZNE) Goettingen, 37077 Göttingen, Germany. [2]Institute of Psychiatric Phenomics and Genomics (IPPG), University Hospital, LMU Munich, Munich, Germany. [3]Institute of Neuroimmunology and Multiple Sclerosis Research, University Medical Center Göttingen, Göttingen, Germany. [4]Department of Cellular Neurology, Hertie Institute for Clinical Brain Research, University of Tübingen, Germany; Germany and German Center for Neurodegenerative Diseases (DZNE), Tübingen, Germany. [5]Department of Psychiatry and Psychotherapy, University Hospital, LMU Munich, Munich, Germany. [6]Department of Psychiatry and Psychotherapy, University Hospital Bonn, Medical Faculty, University of Bonn, Bonn, Germany. [7]Institute of Human Genetics, University Hospital Bonn, Medical Faculty, University of Bonn, Bonn, Germany. [8]Clinic for Psychiatry, Psychotherapy and Psychosomatics, Augsburg University, Medical Faculty, Bezirkskrankenhaus Augsburg, Augsburg 86156, Germany. [9]Department of Psychiatry and Psychotherapeutic Medicine, Research Unit for Bipolar Affective Disorder, Medical University of Graz, Graz 8036, Austria. [10]Department of Psychiatry, Ruhr University Bochum, LWL University Hospital, Bochum 44791, Germany. [11]Department of Pathology, Lifespan Academic Medical Center, Alpert Medical School of Brown University, Providence, RI 02903, USA. [12]Research Group for Genome Dynamics in Brain Diseases, 37077 Göttingen, Germany. [13]Department of Psychiatry and Psychotherapy, University Medical Center Goettingen, 37077 Göttingen, Germany. [14]Cluster of Excellence "Multiscale Bioimaging: from Molecular Machines to Networks of Excitable Cells" (MBExC), University of Göttingen, Göttingen, Germany. [15]These authors contributed equally: Lalit Kaurani, Md Rezaul Islam, Urs Heilbronner. ✉E-mail: lalit.kaurani@dzne.de; thomas.schulze@med.uni-muenchen.de; peter.falkai@med.uni-muenchen.de; fsananb@gwdg.de; a.fischer@mail.gwdg.de

proteins implicated in disease onset and progression (Rupaimoole and Slack, 2017; Damase et al, 2021). Of particular interest are miRs, which are 19-22 nucleotide-long RNA molecules that regulate protein homeostasis via binding to target mRNAs, leading either to their degradation or reduced translation (Gurtan and Sharp, 2013). miRs have been intensively studied as biomarkers and therapeutic targets in cancer (Rupaimoole and Slack, 2017). There is also emerging evidence from genetic studies in humans as well as functional data from mouse models that miRs play a role in CNS diseases including SZ (Fischer, 2014; Martins and Schratt, 2021; Sakamoto and Crowley, 2017; Sargazi et al, 2022). In addition, several studies reported changes in miR levels in blood samples of SZ patients using either qPCR analysis of selected targets or genome-wide approaches. The current findings have been summarized in several review articles (Roy et al, 2020; Ghafouri-Fard et al, 2021; Tsermpini et al, 2022). Despite this progress, there are still only few reports on the function of candidate miRs (Liang et al, 2022). Nevertheless, analysis of miRs in liquid biopsies is highly valuable because one miR can affect many target genes, and thus changes in miR levels can indicate the presence of multiple pathologies (Zampetaki et al, 2012; Fischer, 2014; Condrat et al, 2020). Moreover, miRs also participate in inter-organ communication (Jose, 2015; Bayraktar et al, 2017), suggesting that alterations of miR levels in liquid biopsies may inform about relevant pathological processes in other organs, including the brain. This is important since the analysis of the molecular processes underlying neuropsychiatric diseases in post-mortem human brain tissue is challenging because it might be affected e.g., by peri-mortem events or the timing of postmortem tissue sampling. Furthermore, the onset of the disease often precedes tissue collection by decades. In contrast, liquid biopsies such as blood samples are easy to collect on the premise that molecular changes in blood mirror changes in the brain. In this context, the analysis of the microRNAome in liquid biopsies could be a suitable approach to identify candidate microRNAs that may play a role in the onset and progression of SZ.

In the present study, we performed small RNA sequencing in blood samples of control participants ($n = 331$) and schizophrenia patients ($n = 242$) of the PsyCourse Study (Budde et al, 2018) (http://www.psycourse.de/). By cross-correlating our findings with data from postmortem human brain tissue, we identified miR-99b-5p as a promising biomarker candidate that is decreased in blood and in the prefrontal cortex of SZ patients, and correlates with disease phenotypes. Furthermore, we found decreased levels of miR-99b-5p in the prefrontal cortex of mice to elicit SZ-like phenotypes and activate pathways linked to innate immunity. In line with these observations, inhibition of miR-99b-5p in microglia increased phagocytosis and reduced the number of synapses. Finally, we were able to demonstrate that this effect is controlled by the miR-99b-5p target gene *Zbp1*, an upstream regulator of innate immunity (Kuriakose and Kanneganti, 2018). Taken together, our data suggest that targeting miR-99b-5p or its target *Zbp1* could provide a novel approach towards the treatment of SZ patients.

# Results

## miR-99b-5p levels are decreased in SZ patients

To identify microRNAs that play a role in the pathogenesis of SZ, we conducted small RNA sequencing of blood samples obtained from 573 participants of the PsyCourse Study (http://www.psycourse.de/). We analyzed 331 healthy controls and 242 SZ patients (Figs. 1A and EV1; Dataset EV1; Budde et al, 2018). After performing data normalization and correcting for confounding factors, we decided to employ several exploratory approaches with the aim of identifying microRNAs that play a role in SZ. Firstly, we conducted a Weighted Gene Co-Expression Network Analysis (WGCNA) to investigate the gene expression patterns in the context of schizophrenia (SZ). Our primary objective of this approach was to identify the co-expression modules that exhibited significant differences between groups. Subsequently, we aimed to analyze whether a Gene Ontology (GO) analysis of the targets of microRNAs present within these modules would provide insights into any over-represented biological processes and signaling pathways. This analysis revealed the presence of eight co-expression modules, with three showing significant decreases (Fig. 1B) and five showing increases (Fig. 1C) in SZ patients (see Dataset EV2). Among these modules, those represented by the turquoise, yellow, blue, and red colors demonstrated the most substantial differences ($P < 0.0001$) among the groups. To better understand the biological relevance of these modules, we explored whether the expression levels of miRs within these modules correlated with clinical phenotypes defined by the Positive and Negative Syndrome Rating Scale (PANSS), Beck Depression Inventory (BDI-II), and the Global Assessment of Functioning (GAF) score, all of which are altered SZ patients. Namely, PANSS and BDI-II scores are increased, while the GAF score is decreased in SZ patients (see Fig. EV1).

Out of the eight co-expression modules, the modules represented by turquoise, yellow, blue, and red colors displayed significant correlations with all disease phenotypes. Notably, the module represented by turquoise color exhibited a significant negative correlation with PANSS and BDI-II scores, while it positively correlated with the GAF score (see Fig. 1D). This is consistent with the decreased expression of this module in SZ. Conversely, the modules represented by green, yellow, blue, and red colors positively correlated with PANSS and BDI-II scores and negatively correlated with the GAF score, in line with their increased expression in SZ (see Fig. 1D). The module represented by pink color was negatively correlated with PANSS scores and positively correlated with the GAF, consistent with its decreased expression in SZ patients. This module did not show a correlation with the BDI-II score (see Fig. 1D). In contrast, the modules represented by black and brown colors did not exhibit significant correlations with any of the analyzed phenotypes.

Next, we conducted a GO term analysis on the confirmed targets of these miRs within each module. For all modules, inflammatory processes were overrepresented among the affected pathways, suggesting that the altered microRNA levels may indicate a role of neuroinflammation in SZ, which aligns with the proposed role of neuroinflammation in the pathogenesis of SZ (Buckley 2019; Rodrigues-Neves et al, 2022) (see Appendix Fig. S1; Dataset EV3).

With the aim to further refine the list of candidate microRNAs that may play a role in the pathogenesis of SZ, we analyzed the small RNA-sequencing data from controls and SZ patients via differential expression analysis (Fig. 1E) and detected 59 upregulated and 34 downregulated microRNAs (Dataset EV4).

In addition, we performed small RNA sequencing from the prefrontal cortex of SZ patients ($n = 13$) and controls ($n = 17$). We detected 28 microRNAs that were significantly decreased in SZ patients (FDR < 0.01, log2FC > 1) and 31 that were significantly upregulated (Fig. 1F; Dataset EV5).

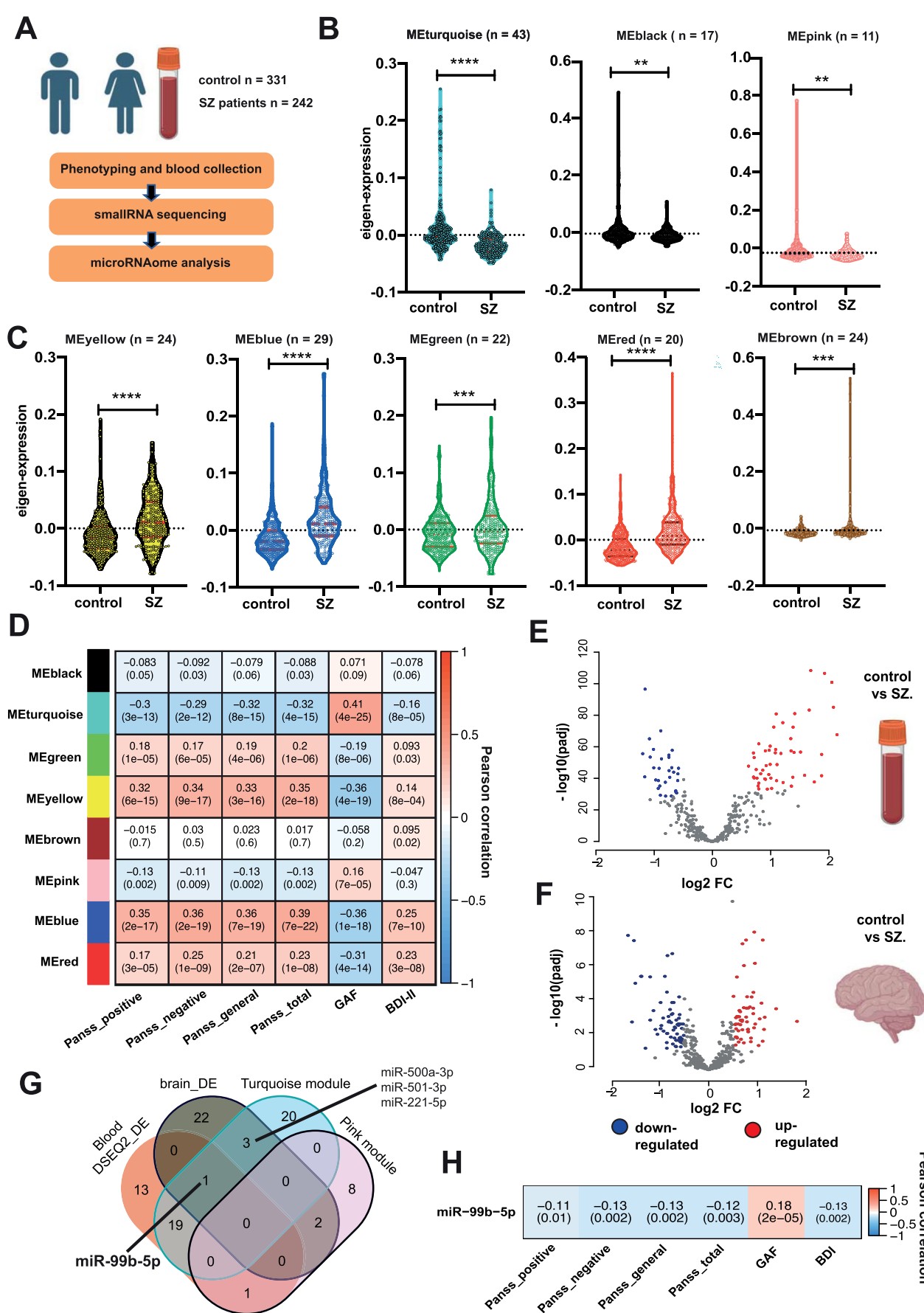

**Figure 1.  Identification of microRNAs that play a role in the pathogenesis of SZ.**

(A) Experimental scheme. (B) Violin plots showing the results of WGCNA analysis. Depicted is the comparison of the eigenexpression in the 3 co-expression modules in SZ patients ($n = 242$) and in control ($n = 331$), showing a decrease in SZ patients. The '$n$' value for each module indicates the number of associated microRNAs. (C) Violin plots showing the results of WGCNA analysis. The eigenexpression of the 5 co-expression modules was higher in SZ patients than in controls. The '$n$' value for each module indicates the number of associated microRNAs (for (B) and (C): unpaired t test; **$P < 0.01$, ***$P < 0.001$; ****$P < 0.0001$, a $P$ value $< 0.01$ was considered as significant). (D) Heat map showing the correlation of the eigenexpression of the co-expression modules shown in (A) and (B) with the corresponding clinical phenotypes. The numbers in each rectangle represent the correlation (upper number) and the corresponding $p$-value (lower number). A $P$ value $< 0.01$ was considered as significant. (E) Volcano plot depicting the results of the differential expression analysis when comparing miRs in blood samples from SZ patients ($n = 242$) and controls ($n = 331$) shown in (A) (FDR $< 0.01$, log2FC $> 1$). (F) Volcano plot demonstrating the results of the differential expression analysis when comparing miRs from postmortem brain samples from SZ patients ($n = 13$) and controls ($n = 17$) (FDR $< 0.01$, log2FC $> 1$). (G) Venn diagram comparing the microRNAs detected in blood samples when performing differential expression analysis (Blood DESEQ2_DE), the microRNAs of the ME_Turquoise and ME_Pink co-expression modules and the microRNAs differentially expressed when comparing postmortem brain tissue (brain_DE). miR-99b-5p is the only microRNA decreased in all comparisons. (H) Heat map showing the correlation of miR-99b-5p expression levels to the clinical phenotypes for the individuals as analyzed in (A). The numbers in each rectangle represent the correlation (upper number) and the corresponding $p$-value (lower number). Data information: In the violin plots in (B, C), the body illustrates the distribution of the data, showing both density and range. The centerline indicates the median, while the thickness of the plot represents the density of the data at different values. Source data are available online for this figure.

Next, we examined whether any of these microRNAs are also found among the differentially expressed microRNAs that were altered in blood samples when compared via differential expression analysis, and within the co-expression modules decreased in SZ patients. First we looked at the upregulated microRNAs. Three miRs of the co-expression module represented by the yellow color and one miR of the co-expression modules represented by the blue and green colors were significantly increased in the brain and in the blood when analyzed via differential expression analysis. These were miR-101-3p, miR-378a-3p, miR-21-5p, miR-192-5p and miR-103a-3p (Fig. EV2A). MiR- 21-3p has been associated with SZ while the other 4 miRs have been studied in the context of other neuropsychiatric or neurodegenerative diseases (Dataset EV6). Next, we analyzed the downregulated miRs and the co-expression modules that were decreased in SZ patients. We found four miRs, namely miR-500a-3p, miR-501-3p, miR-221-5p, and miR-99b-5p, that were part of the decreased co-expression module represented by the turquoise color and were also decreased in the postmortem brain of SZ patients (Fig. 1G). Of these 4 microRNAs miR-501-3p has been recently linked to schizophrenia (Liang et al, 2022; Dataset EV6). MiR-99b-5p was the only candidate that was (1) part of a significantly downregulated co-expression module, (2) was significantly downregulated in the brain and (3) blood of SZ patients when analyzed via differential expression analysis.

In summary, our data reveal a number of interesting candidate miRs such as miR-501-3p and miR-21-5p that have been already linked to SZ in previous studies (Liu et al, 2017; Liang et al, 2022). Most of the other miRs have been detected in the context of other brain diseases including Alzheimer's disease (AD), Major depression (MD) or Amyotrophic lateral sclerosis (ALS) (Dataset EV6). Except for miR-500a-3p and miR-221-5p, the expression of all other candidate miRs is significantly correlated to the PANSSs, GAF, and BDI-II scores (Figs. 1H and EV2B,C). While all of these candidate miRs would warrant further functional analysis in the context of SZ, we decided to focus on miR-99b-5p since it has not been linked to any brain disease yet and comparatively little is known about this miR in general.

## Decreasing miR-99b-5p leads to SZ-like phenotypes in mice and the upregulation of genes linked to inflammatory processes

The role of miR-99b-5p in the brain has not been intensively studied and thus no data are available in the context of

neuropsychiatric diseases such as SZ, making it a novel candidate in need of further evaluation. Before performing mechanistic studies, we decided to employ mice in a model system to test the hypothesis that inhibition of miR-99b-5p could lead to behavioral alterations in mice, including SZ-like phenotypes such as pre-pulse inhibition of the startle response (PPI). Therefore, we generated lipid nanoparticles (LNPs) containing miR-99b-5p inhibitors (anti-miR99b). First, we confirmed target engagement of anti-miR-99b in cultures cells (Fig. EV3A,B). Subsequently, we injected anti-miR99b into the prefrontal cortex (PFC) of mice. Corresponding oligonucleotides with a scrambled nucleotide sequence were used as controls (sc-control). We could detect significantly lower miR-99b-5p levels in the PFC when measured 5 or 10 days after the injections via qPCR (Fig. 2A). MiR-99b-5p levels were not affected when qPCR was performed from hippocampal tissue of injected mice (Fig. EV3C). To test behavioral alterations, we injected either anti-miR-99b or sc-control to the PFC of mice. Explorative behavior measured in the open field test was similar in all groups (Fig. 2B). However, anti-miR-99b-treated mice spent less time in the center of the open field, which is indicative of increased anxiety (Fig. 2C), a phenotype commonly observed in schizophrenia patients (Achim et al, 2011). We also analyzed anxiety behavior in the elevated plus maze test. Anti-miR-99b-treated mice spent less time in the open arms, which indicates increased anxiety (Fig. 2D). Another valid animal model to test schizophrenia-like behavior in rodents is PPI which is used to measure sensory-gating function (Swerdlow et al, 1994). PPI is impaired in SZ patients, can easily be assayed in mice and is impaired in mouse models for SZ (Light and Braff, 1999; van den Buusse, 2010). We observed that mice injected with anti-miR-99b displayed significantly impaired PPI responses when compared to the sc-control group (Fig. 2E). The basic startle response was unaffected (Fig. 2F), suggesting that decreasing the levels of miR-99b-5p in the PFC of mice can lead to SZ-like phenotypes. Although these data are encouraging, it has to be mentioned that mice cannot fully recapitulate the complex phenotypes associated with schizophrenia in humans. Thus, our results provide further evidence for a role of miR-99b-5p in schizophrenia but cannot fully establish a causal relationship.

To gain first insights into the molecular processes controlled by miR-99b-5p in the brain, we injected another group of mice with anti-miR-99b and sc-control oligonucleotides and isolated PFC tissue 5 days later for RNA sequencing analysis. Differential expression analysis revealed 147 deregulated genes (adjusted $p$-value $< 0.1$, log2FC $+/- 0.2$), of which 113 genes were upregulated

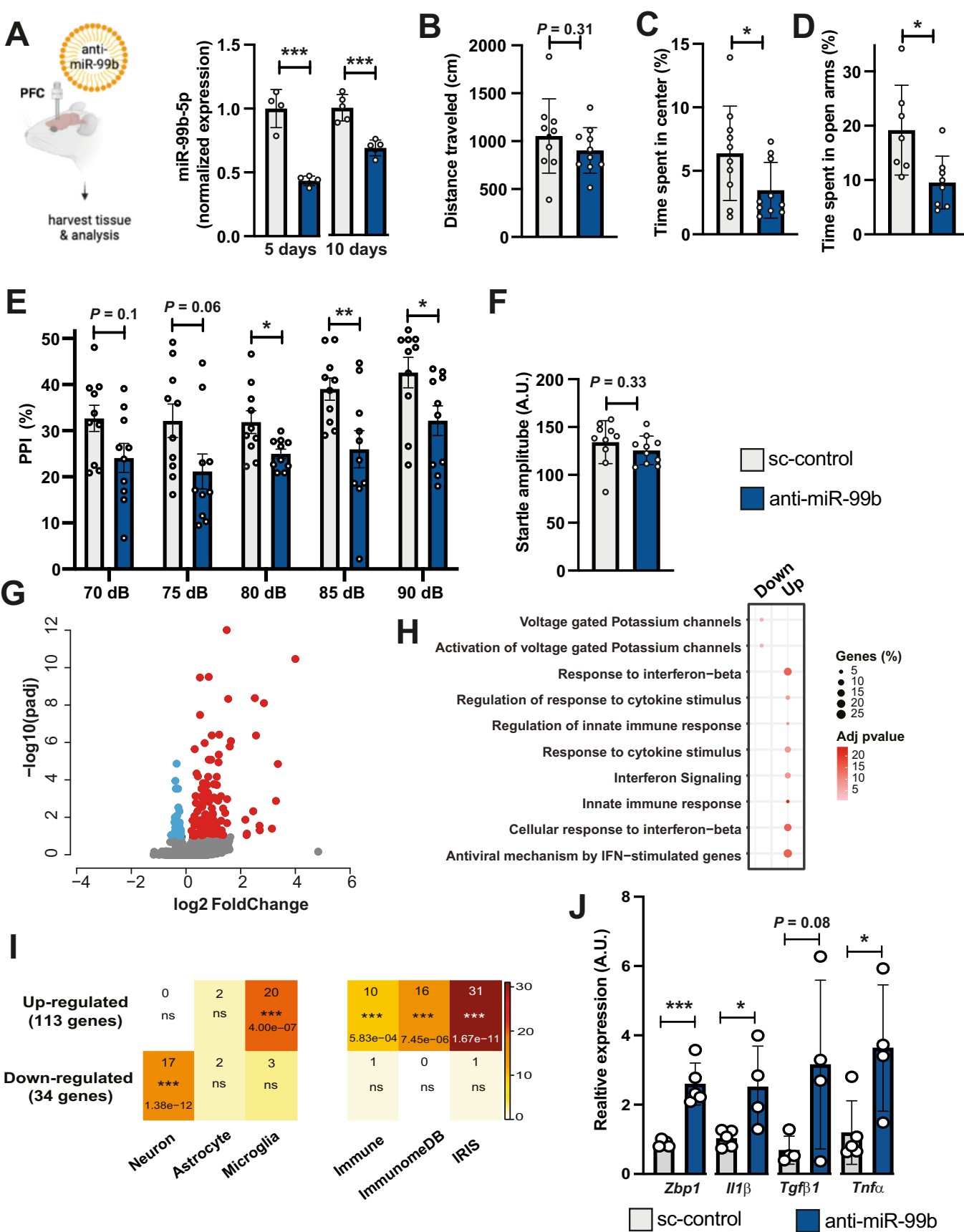

◀

**Figure 2. Decreasing miR-99b-5p levels in the PFC of mice leads to SZ-like phenotypes and increases the expression of genes linked to microglia activation.**

(A) Left panel: Experimental design. Right panel: Bar graph showing qPCR results for miR-99b-5p in tissue obtained from the PCF of mice 5 or 10 days after injection of anti-miR-99b or sc-control oligonucleotides ($n = 4$/group; ***$P < 0.001$, unpaired t test, Error bars indicate SD). (B) Bar graph showing the distance traveled in the open field test of mice injected to the PFC with either anti-miR-99b or sc-control oligonucleotides ($n = 10$/group; unpaired t test; Error bars indicate SD). (C) Bar graph showing the time spent in the center of the open field in mice injected to the PFC with either anti-miR-99b or sc-control oligonucleotides ($n = 10$/group; *$P < 0.05$; unpaired t test; Error bars indicate SD). (D) Bar graph showing the time spent in the open arms when an elevated plus maze test was performed in mice injected to the PFC with either anti-miR-99b or sc-control oligonucleotides ($n = 10$/group; *$P < 0.05$; unpaired t test; Error bars indicate SD). (E) Bar graph showing the results of a PPI experiment of mice injected with either anti-miR-99b or sc-control oligonucleotides. A repeated-measures ANOVA was run to test whether anti-mir-99b had any effects on PPI at different intensity levels. Given that we observed a statistically significant effect, F(2.6,47.8) = 5.7, *$P < 0.05$, we conclude that PPI is affected in anti-miR-99b injected mice. To further reinforce this finding, the panel further shows post hoc unpaired t-tests applied independently at each indicated dB level ($n = 10$/group); *$P < 0.05$; **$P < 0.01$; Error bars indicate SD (F). Bar graph showing the basic startle response among groups ($n = 10$/group; unpaired t test; Error bars indicate SD). (G) Volcano plot showing the differentially expressed genes (upregulated in red, downregulated in blue) when RNA-seq was performed from the PFC of mice injected with either anti-miR-99b or sc-control oligonucleotides. Genes with log2-fold change ± 0.5 and adjusted $p$ value < 0.05 are highlighted ($n = 3$/group, Wald test:Deseq2). RNA seq was performed from individual mice. (H) GO-term analysis of the upregulated genes found in (E). (I) Heat maps showing the enrichment of the upregulated genes as determined in (E) in various datasets. Left panel shows that the upregulated genes are enriched for microglia-specific genes, while the downregulated genes are enriched for neuron-specific genes. The right panel shows that the upregulated genes are over-represented in 3 different databases for immune function-related genes (Fisher's exact test). (J) Bar graph showing the qPCR results of the *Zbp1*, *Il1ß*, *Tgfb1*, and *Tnfa* genes in FACS-sorted microglia collected from the PFC of mice injected with anti-miR-99b or sc-control oligonucleotides ($n = 4$ or 5/group; unpaired t test; *$P < 0.05$; ***$P < 0.001$). Error bars indicate SD. Data information: Bars and error bars in panels (B–F, J) indicate mean ± SD. Source data are available online for this figure.

and 34 were downregulated (Fig. 2G; Dataset EV7). Gene ontology (GO) analysis identified several significant processes for the upregulated genes and revealed that these genes were mainly linked to processes such as innate immunity and interferon signaling (Fig. 2H; Dataset EV8). GO analysis of the downregulated genes revealed 2 significant processes that are "voltage-gated potassium channels" and "Activation of voltage-gated Potassium channels" (Fig. 2H; Dataset EV8). These data suggest that miR-99b-5p in the PFC may regulate mRNAs linked to immune-related processes. To further test this hypothesis, we compared the list of upregulated genes with gene expression data from neurons, astrocytes, and microglia as well as genes present within 3 different immune function-related gene expression databases. We observed that the upregulated genes were highly enriched in microglia ($P = 4 \times 10^{-12}$), while no enrichment was observed in astrocytes or neurons. The downregulated genes were significantly enriched in neurons ($P = 1.38 \times 10^{-12}$). In line with this, the upregulated genes were significantly overrepresented in 3 databases for genes linked to immune function, namely the immune, immunome, and IRIS databases (Fig. 2I). Together, these data suggest that the levels of miR-99b-5p in the PFC of mice may specifically increase the expression of immune-related genes in microglia. To further test this we administered anti-miR-99b or sc-control to the PFC of mice and subsequently isolated CD45^low/CD11b^+ microglial cells via fluorescence-activated cell sorting (FACS). While microglia cell numbers did not differ between groups, miR-99b-5p expression was significantly decreased in microglia isolated from anti-miR-99b-treated mice (Appendix Fig. S2). Next, we tested the expression of selected key pro-inflammatory genes that were significantly increased in the analysis of bulk tissue such as *Zbp1*. Increased expression of *Zbp1* was confirmed in sorted microglia cells (Fig. 2J). In addition, we hypothesized that pro-inflammatory genes that were increased in the bulk data but failed to reach statistical significance might exhibit a more pronounced and statistically significant alteration in expression when specifically assessed within the microglial population. Such were for example *Il1ß* ($P = 0.1$ in bulk tissue), *Tgfb1* ($P = 0.11$ in bulk tissue), and *Tnfa* ($P = 0.16$ in bulk tissue). Except for *Tgfb1* ($P = 0.08$), significantly increased expression was observed in microglia obtained from anti-miR-99b-treated mice (Fig. 2J).

These data suggest that miR-99b-5p controls microglia-mediated immunity in the PFC, which is in agreement with previous studies linking aberrant microglia function and neuroinflammation to the pathogenesis of SZ (Bayer et al, 1999; Zhuo et al, 2023; Breitmeyer et al, 2023).

## miR-99b-5p controls microglia-mediated immune function and affects dendritic spine number

To further explore the role of miR-99b-5p in microglia, we cultured primary microglia from the cortex of mice and treated these cells with anti-miR-99b or sc-control LNAs followed by RNA sequencing (Fig. 3A). Differential expression analysis revealed 139 deregulated genes, of which 104 were up- and 35 were downregulated (Fig. 3B; Dataset EV9). GO-term and KEGG-pathway analysis revealed that the upregulated genes were linked to immune activation and phagocytosis (Fig. 3C; Dataset EV10). In agreement with the in vivo data, we observed an increased expression of *Il1ß*, *Tgfb1, and Tnfα*, which could be confirmed via qPCR (Fig. 3D). Next, we investigated phagocytosis, a key function of microglia that is altered during neuroinflammation (Borst et al, 2021). In a first approach we employed immortalized microglia (IMG) cells. Similar to the treatment with the lipopolysaccharide (LPS) commonly used to induce microglia activation, inhibition of miR-99b-5p caused an upregulation of pro-inflammatory cytokines and increased phagocytosis as measured via the uptake of fluorescent latex beads (Fig. EV4). Encouraged by these data we performed similar experiments in microglia isolated from the mouse PFC. In line with the data obtained in IMG cells, treatment of microglia with anti-miR-99b LNAs significantly increased phagocytosis (Fig. 3E).

Aberrant microglia activity can have detrimental effects on neuronal plasticity (Sellgren et al, 2019). To test whether the reduced expression of miR-99b-5p in microglia could affect neuronal plasticity, we performed a co-culturing experiment. Microglia were first treated with sc-control or anti-miR-99b LNAs for 48 h before being harvested and transferred to cortical neuronal cultures (Fig. 3F). RNA was isolated from these co-cultures after 48 h and subjected to RNA-seq. Differential expression analysis revealed 366 deregulated genes (155 upregulated and 211 downregulated genes, adjusted $p$ value < 0.05, log2FC +/− 0.1); Fig. 3G;

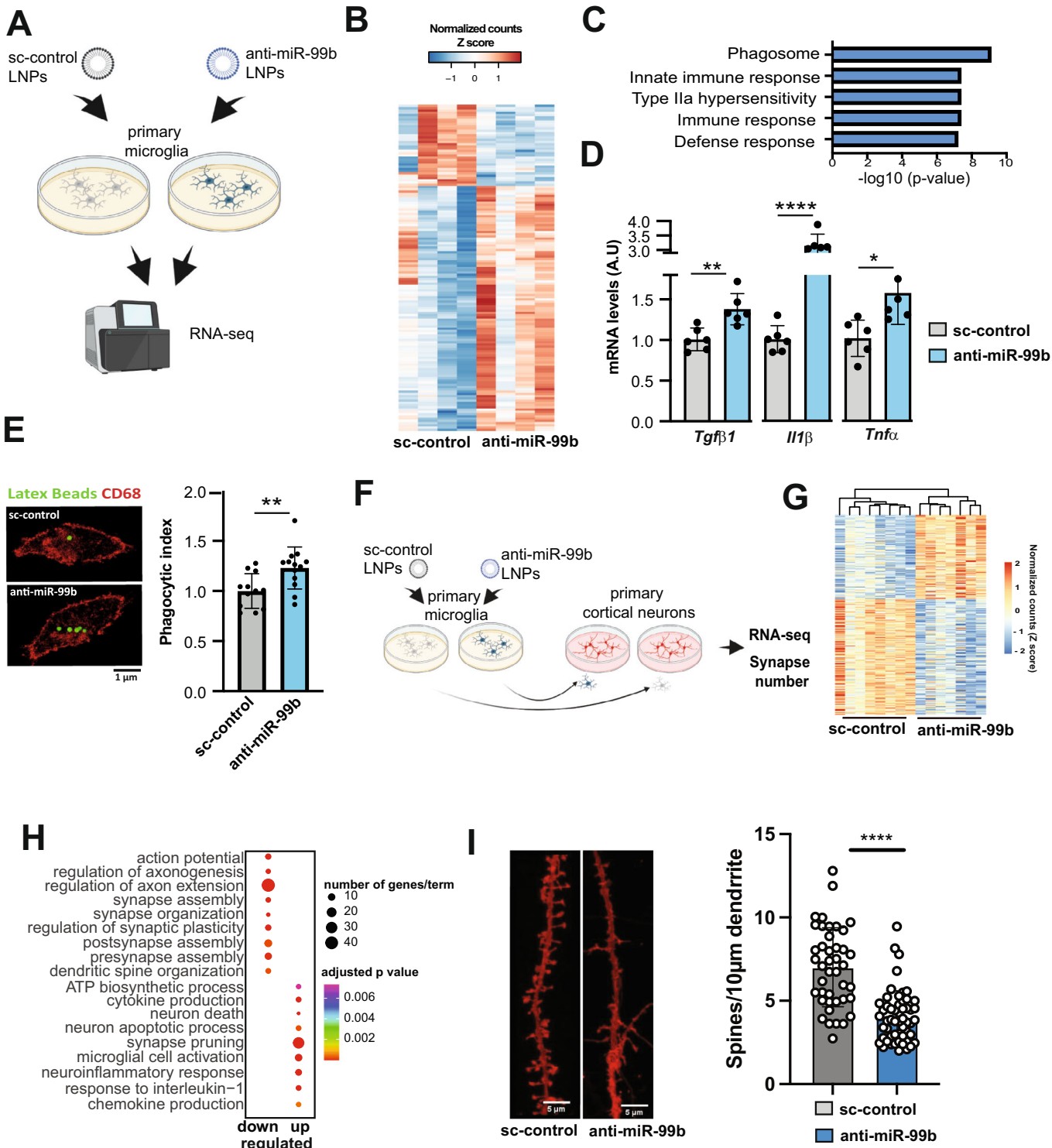

Dataset EV11). GO term analysis of the upregulated genes revealed neuroinflammatory processes, neuron death, neuron apoptotic processes as well as synaptic pruning (Fig. 3H; Dataset EV12), while downregulated genes were associated with processes indicating loss of synaptic function such as regulation of axon extension or synapse organization (Fig. 3H; Dataset EV13).

These data are in agreement with our previous findings suggesting that loss of miR-99b-5p increases inflammatory processes in microglia. It is particularly interesting that synaptic pruning is detected as a major process increased in the co-cultures, since synaptic pruning has been linked to SZ (Sellgren et al, 2019). In line with this, key factors of the complement system known to

**Figure 3.  Decreasing miR-99b-5p levels in microglia increases phagocytosis and reduces synapse number in cortical neurons.**

(A) Left panel: Experimental design. (B) Volcano plot showing differential expressed genes when comparing microglia treated with anti-miR-99b or sc-control LNAs. Genes with statistical significance are highlighted. (C) Bar chart showing the top GO terms represented by the upregulated genes shown in (B) (ClueGO v2.2.5 plugin of Cytoscape 3.2.1 was used. Two-sided hypergeometric test was used to calculate the importance of each term and the Benjamini-Hochberg procedure was applied for the $P$ value correction.). (D) Bar charts showing qPCR results for *Tgfb1*, *Il1ß*, and *Tnfa* comparing microglia treated with anti-miR-99b or sc-control LNAs (n = 6/group; unpaired t test; *$P$ < 0.05; **$P$ < 0.01, ****$P$ < 0.0001; Error bars indicate SEM). (E) Bar chart showing the results of a phagocytosis assay performed in microglia treated with anti-miR-99b in comparison to cells treated with sc-control LNAs. The percentage of phagocytic index represents (# of total engulfed beads in an image/# of total cells identified in an image; n = 13 independent experiments; unpaired t test; **$P$ < 0.01; Error bars indicate SEM). (F) Experimental scheme illustrating the co-culture experiment. (G) Heat map showing the differentially expressed genes from the experiment described in (F). (H) Plot showing the results of a GO term analysis for the up- and downregulated genes displayed in (G) (ClueGO v2.2.5 plugin of Cytoscape 3.2.1 was used. Two-sided hypergeometric test was used to calculate the importance of each term and the Benjamini-Hochberg procedure was applied for the $P$ value correction.). (I) Left panel: Representative image showing DIL dye staining to visualize dendritic spines in co-cultures as illustrated in (F). Scale bar 5 μm. Right panel: Bar chart showing the statistical quantification (spines/10 μm dendrite) of the data depicted in (I). Data was analyzed using *t*Test. ****$P$ < 0.0001). Error bars indicate SD. Data information: Bars and error bars in panels (D, E, I) indicate mean ± SEM. Source data are available online for this figure.

drive pathological synaptic pruning were increased in microglia treated with anti-miR-99b, as well as in the corresponding co-cultures and also in the postmortem human prefrontal cortex of schizophrenia patients (Fig. EV5). When we analyzed the number of dendritic spines, we observed that spine density was significantly reduced in neurons co-cultured with microglia that had received anti-miR-99b, when compared to cultures treated with corresponding control microglia (Fig. 3I).

## miR99b-5p control neuroinflammation via the regulation of Zbp1

The three RNAseq datasets obtained from the PFC of mice, primary microglia and the co-cultures consistently show that knockdown of miR-99b-5p increases the expression of genes linked to inflammatory processes. Many of the gene expression changes likely represent secondary effects. To better understand the mechanisms by which miR-99b-5p controls neuroinflammation, we aimed to identify direct targets of miR-99b-5p (Dataset EV14). When we analyzed the RNA-seq data obtained from the PFC (see Fig. 2), we identified 13 out of 113 genes as potential mRNA targets of miR99b-5p when using miRwalk as a prediction tool (Fig. 4A). GO term analysis was performed for the 13 genes and revealed that they are linked to inflammatory processes including type I interferon signaling (Appendix Fig. S3; Dataset EV15), These processes have been previously linked to schizophrenia (Müller et al, 2015; van Mierlo et al, 2020). Seven of these genes were also upregulated in primary microglia treated with anti-miR-99b, and among them were key regulators of inflammatory processes such as *Stat1* which was found to be hyperactive in blood samples of SZ patients (Sharma et al, 2016). A gene that specifically caught our attention was *Zbp1* because the corresponding protein - also known as the DNA-dependent activator of interferon regulatory factors (*Dai*) - is a key regulator of pro-inflammatory processes that result in the activation of inflammatory caspases and the induction of *Il1ß* (Kuriakose and Kanneganti, 2018). Thus, *Zbp1* represented a rather upstream factor in the inflammatory cascade. On this basis, we speculated that the regulation of *Zbp1* could be a key mechanism by which miR-99b-5p regulates inflammatory processes and may contribute to the pathogenesis of SZ when it is increased. It should be mentioned that *Zbp1* is not identified as a predicted target of miR-99b-5p when using the tools such Targetscan or miRDB for prediction.

Since target prediction can only be a first hint to the function of a microRNA, we performed a luciferase assay to directly test the regulation of *Zbp1* by miR-99b-5p. We used the renilla dual-luciferase reporter vector harboring the *Zbp1*-3′UTR. Co-transfection of this vector with miR-99b-5p mimic significantly reduced the luciferase activity (Fig. 4B), but this was not the case when scramble control was used or when the miR-99b-5p seed region was mutated (Fig. 4B). Moreover, we observed that ZBP1 protein levels were significantly increased in primary microglia treated with anti-miR-99b (Fig. 4C). We also confirmed that Zbp1 and miR-99b-5p are expressed in microglia in vivo. QPcr analysis revealed that in the mouse PFC, miR-99b-5p is highly expressed in microglia but also in astrocytes and oligodendrocytes, while *Zbp1* is almost exclusively detected in microglia (Appendix Fig. S4). In sum, these data suggest that miR-99b-5p can directly regulate *Zbp1* levels in microglia.

On this basis we decided to investigate whether the inflammatory phenotypes induced in response to decreased miR-99b-5p levels depend on *Zbp1*. So far, we have found that decreased levels of miR-99b-5p lead to enhanced phagocytosis and increased expression of pro-inflammatory cytokines such as *Il1ß* which has been associated with *ZBP1* activity (Muendlein et al, 2021). Another important step in ZBP1-mediated orchestration of inflammation is the activation of pro-inflammatory caspases (Shao et al, 2022; de Reuver et al, 2022), and therefore we examined whether reduced miR-99b-5p levels would also affect the activity of pro-inflammatory caspases. Indeed, when primary microglia were treated with anti-miR-99b, caspase activity was significantly increased when compared to that in cells treated with sc-control LNAs (Fig. 4D). Similar findings were obtained when protein lysates isolated from the PFC of mice injected with either anti-miR-99b or sc-control LNAs were analyzed for caspase activity (Fig. 4E).

To test whether the miR-99b-5p-mediated increase in caspase activity, *IL-1ß* expression and phagocytosis depends on *Zbp1*, we treated primary microglia with either anti-miR-99b alone or in combination with an anti-sense oligonucleotide (ASO) targeting *Zbp1* (Zbp1-ASO). Prior to this we confirmed the efficacy of *Zbp1*-ASO (Appendix Fig. S5). In agreement with our previous observation, anti-miR-99b treatment increased caspase activity. This effect was ameliorated in microglia treated with anti-miR-99b and *Zbp1*-ASO (Fig. 4F). Similar observations were made when we analyzed *IL1ß* expression (Fig. 4G) and phagocytosis (Fig. 4H). These data suggest that ZBP-1 plays an important role in mediating the neuroinflammatory processes downstream of miR-99b-5p.

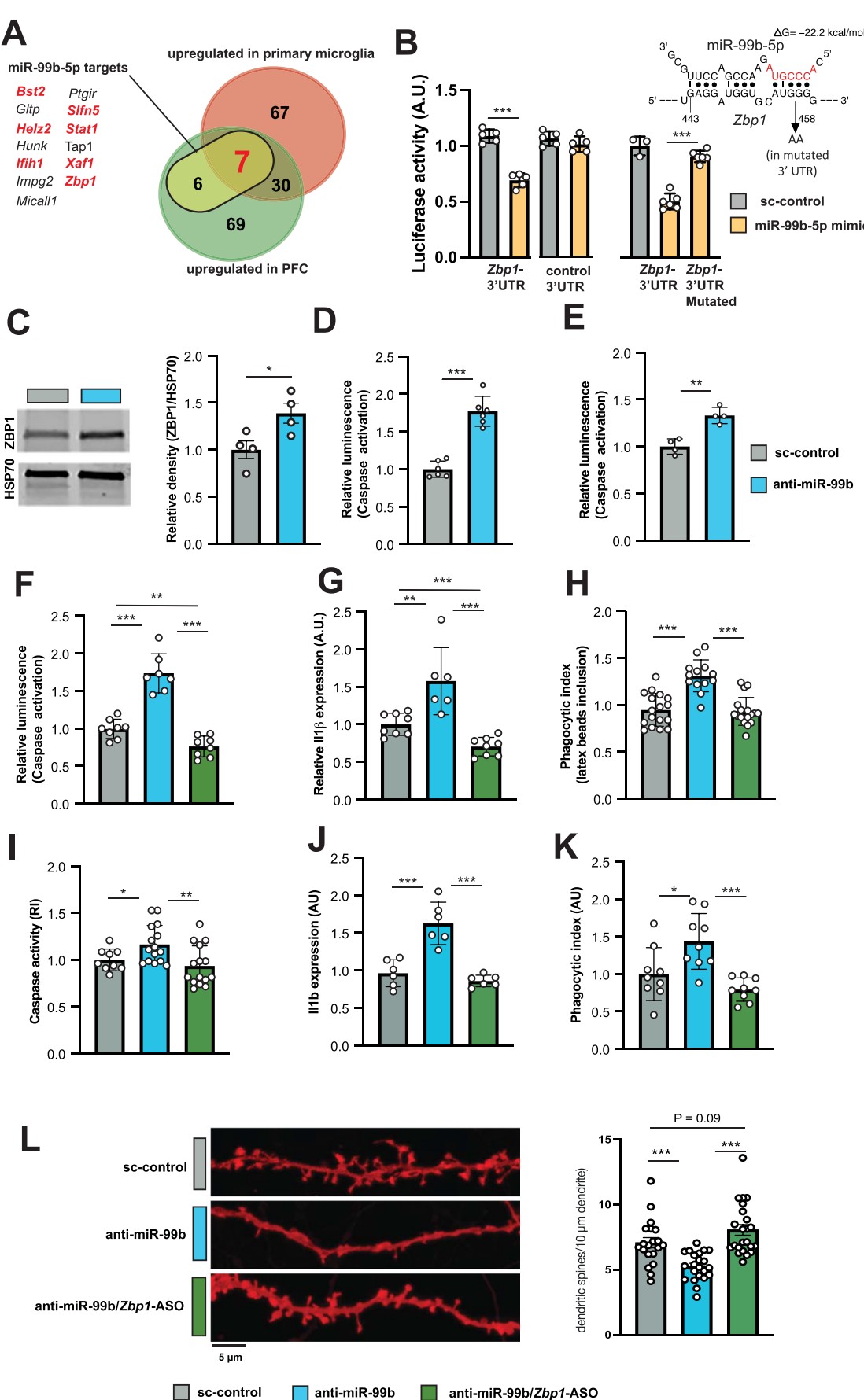

**Figure 4.   miR-99b-5p regulates neuroinflammatory phenotypes via *Zbp1*.**

(A) Venn diagram comparing the genes upregulated in the PFC of mice and in primary microglia when injected or treated with anti-miR99b vs. sc-control LNAs, respectively. The data is further compared to the identified 13 miR-99b-5p target mRNAs detected in the PFC dataset. The left panel shows the gene names of the 13 miR-99b-5p target mRNAs. Red indicates miR-99b-5p targets upregulated in the PCF and in primary microglia upon anti-miR-99b treatment. (B) Bar graphs showing the results of the luciferase assay. Left panel: In comparison to sc-control LNAs, administration of miR-99b-5p mimic decreases luciferase activity when cells express the Zbp1-3'UTR. This effect is not observed when a control 3'UTR that does not bind miR-99b-5p is used ($n = 6$/group; t-Test; ***$P < 0.001$). Right panel: Similarly, the inhibitory effect of miR99b-5p mimic is not observed when we mutated the miR-99b-5p seed region within the 3' UTR of *Zbp1*. One-way ANOVA revealed a significant difference among the groups ($p < 0.0001$). Subsequent pairwise t-tests were performed to examine specific group differences. The upper right panel shows the predicted binding of miR-99b-5p to the 3'UTR of *Zbp1*. The mutation GG → AA with respect to the data shown in the right panel is indicated by an arrow. (C) Left panel: Representative immunoblot image showing ZBP1 levels in microglia treated with sc-control LNAs or anti-miR-99b. HSP70 was used as a loading control. Right panel: Bar graph showing the quantification of the data depicted in the left panel ($n = 4$/group; unpaired t test; *$P < 0.05$). (D) Bar graph showing quantification of caspase activity in primary microglia treated with sc-control LNAs or anti-miR-99b ($n = 6$/group; unpaired t test; ***$P < 0.001$). (E) Bar graph showing quantification of caspase activity in protein lysates isolated from the PFC of mice injected with anti-miR-99b or sc-control ($n = 4$/group; unpaired t test; **$P < 0.01$). (F) Bar graph showing quantification of caspase activity in primary microglia treated with either sc-control LNAs, anti-miR-99b or anti-miR-99b together with *Zbp1*-ASOs ($n = 6$/group; unpaired t test; **$P < 0.01$; ***$P < 0.001$). (G) Bar graph showing qPCR results for *Il1β* in primary microglia treated with either sc-control LNAs, anti-miR-99b or anti-miR-99b together with *Zbp1*-ASOs ($n = 6$/group; unpaired t test; **$P < 0.01$; ***$P < 0.001$). (H) Bar graph showing the results of a phagocytosis assay performed in primary microglia treated with either sc-control LNAs, anti-miR-99b or anti-miR-99b together with *Zbp1*-ASOs ($n = 16$ independent experiments; unpaired t test; ***$P < 0.001$). (I) Bar graph showing quantification of caspase activity in human iPSC-derived microglia treated with either sc-control LNAs, anti-miR-99b or anti-miR-99b together with *Zbp1*-ASOs ($n = 13$-16 samples/group; unpaired t test; *$P < 0.05$; **$P < 0.01$). (J) Bar graph showing qPCR results for IL1β in human iPSC-derived microglia treated with either sc-control LNAs, anti-miR-99b or anti-miR-99b together with *Zbp1*-ASOs ($n = 6$/group; unpaired t test; ***$P < 0.001$). (K) Bar graph showing the results of a phagocytosis assay performed in human iPSC-derived microglia treated with either sc-control LNAs, anti-miR-99b or anti-miR-99b together with *Zbp1*-ASOs. The percentage of phagocytic index represents (# of total engulfed beads in an image/# of total cells identified in an image; $n = 9$ independent experiments; unpaired t test; *$P < 0.05$; ***$P < 0.001$). (L) Left panel: Representative image showing DIL dye staining to visualize dendritic spines in co-cultures. Scale bar 5 μm. Right panel: Bar chart showing the statistical quantification (spines/10 μm dendrite). One-way ANOVA revealed a difference amongst groups ($P < 0.09$). (The data was further analyzed via tTest; ***$P < 0.001$). In all panels of this figure, error bars represent the Standard Deviation (SD) of the data. RI: relative immunofluorescent. Data information: Bars and error bars in panels (B–L) indicate mean ± SEM. For the data shown in panels (F–L). One-way ANOVA revealed a significant difference among the groups ($P < 0.0001$). Subsequent pairwise t-tests were performed to examine specific group differences. Source data are available online for this figure.

We performed parallel experiments in human iPSC-derived microglia. Similar to the mouse data, administration of anti-miR-99b increased caspase activity (Fig. 4I), IL-1ß expression (Fig. 4J), and phagocytosis (Fig. 4K) when compared to human iPSC-derived microglia treated with sc-control LNAs. These effects were attenuated when anti-miR-99b LNAs were co-administered with *Zbp1*-ASOs (Fig. 4I–K). These data suggest that in human microglia, miR-99b-5p also controls neuroinflammatory processes via the regulation of *Zbp-1* expression. In line with this interpretation, *IL1ß* and *Zbp1* mRNA levels were increased in postmortem human brain samples from SZ patients and controls (Appendix Fig. S6). Administration of anti-miR-99b to astrocytes reduced spine density in co-cultured neurons (see Fig. 3). We repeated this experiment but included an additional group in which astrocytes were treated anti-miR-99b and *Zbp1*-Asos. Our data show that administration of *Zbp1*-ASOs could reverse anti-miR-99b-mediated decrease in spine density (Fig. 4L).

To determine whether knockdown of *Zbp1* would also mitigate the effect of anti-miR-99b treatment on SZ-like behavior in mice, we injected either anti-miR-99b alone or in combination with *Zbp1*-ASOs into the PFC of mice before subjecting the animals to behavior testing. Mice injected with sc-control LNAs served as controls. We confirmed the effect of anti-miR-99b and *Zbp1*-ASO injection into the PFC (Fig. EV3D) and show that PFC injection of do not affect levels in the hippocampus, suggesting brain-region specific actions of the treatment (see Fig. EV3E). The corresponding data revealed that *Zbp1* knockdown rescues anti-miR-99b-mediated impairment of PPI (Fig. 5A,B).

Our findings suggest that reduced miR-99b-5p levels in microglia contribute to the pathogenesis of schizophrenia via the regulation of *Zbp1*-controlled neuroinflammation. Therefore, miR-99b-5p may constitute a novel biomarker for SZ, while targeting miR-99b-5p and/or ZBP1 might represent an effective SZ treatment.

## Discussion

In this study, we combined the analysis of blood samples and postmortem brain tissue to identify miRs involved in the pathogenesis of SZ. Using WGCNA as well as differential expression analysis in blood samples, we identified several miRs that differed between patients and controls and were significantly correlated with SZ phenotypes. GO term analysis of the confirmed target genes of these miRs hinted at a number of molecular processes of which pathways linked to immune function were overrepresented. Such a GO term analysis based on miR target genes is, of course, not ultimately conclusive but our observation is in agreement with previous studies showing that neuroinflammation plays a role in the pathogenesis of SZ (Buckley 2019; Rodrigues-Neves et al, 2022). To further refine the identification of miRs linked to SZ we also performed a differential expression analysis of the small RNA seq data obtained from blood as well as from postmortem brain tissue of SZ patients and controls. We eventually identified nine candidate miRs, miR-101-3p, miR-378a-3p, miR-21-5p, miR-192-5p and miR-103a-3p, miR-500a-3p, mIR-501-3p, miR-221-5p and miR-99b-5p. Except for miR-99b-5p, all of these miRs have been implicated in brain diseases and neuronal plasticity (Dong et al, 2021; Biselli et al, 2022; Zhang et al, 2021; Banach et al, 2022; Barbato et al, 2020; Mundalil Vasu et al, 2014; Zadehbagheri et al, 2019; Maffioletti et al, 2016; Qin et al, 2022; Yoshino et al, 2021; Tang et al, 2019; Van der Auwera et al, 2019). (Huang et al, 2021) and even specifically with SZ (Chen et al, 2016; Liu et al, 2017). Of particular interest is a recent study that found miR-501-3p to be decreased in blood samples of monozygotic twins discordant for SZ (Liang et al, 2022). These data are in agreement with our observation that miR-501-3p was decreased in SZ patients of the PsyCourse study as well as in the postmortem brains of SZ patients.

These data support the validity of our approach and we decided to study miR-99b-5p at the functional level. Inhibiting miR-99b-5p in the prefrontal cortex of mice led to impaired PPI and increased

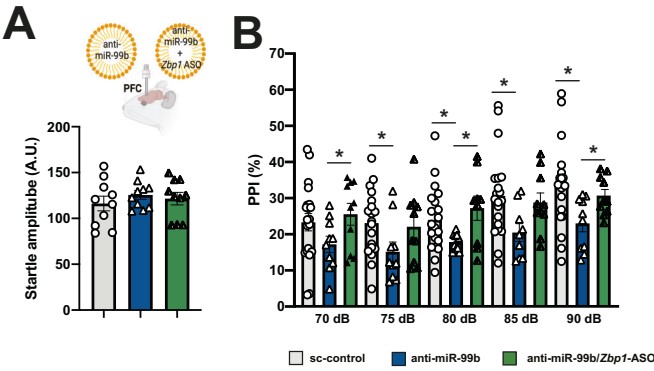

**Figure 5. miR-99b-5p regulates neuroinflammatory phenotypes via *Zbp1* a.**

(A) Upper panel: Experimental design. LNPs loaded with either sc-control, anti-miR-99b, or anti-miR-99b together with *Zbp1* ASOs were injected into the PFC before mice were subjected to behavioral testing. Lower panel: Bar graph showing that the basic startle response is not different amongst experimental groups (n = 10/group). (B) Bar graph showing results from the PPI experiment comparing mice injected to the PFC with either sc-control (n = 10), anti-miR-99b (n = 10), or anti-miR-99b together with *Zbp1* ASOs (n = 10). Missing data made us rely on a linear mixed-model to test whether anti-mir-99b (or anti-mir-99b/Zbp1-ASO) had any effects on PPI. Results indicated either of the injections —or both—had a statistically significant effect, F(2.4,66) = 11.9, *P < 0.0001, but without specifically indicating which one was behind this result. Applying pairwise unpaired *t*-tests to compare each of the groups at specific dB values, we observe that, in agreement with our previous data, PPI is impaired when comparing the sc-control group to mice injected with anti-miR-99b (*P < 0.01 for each intensity ranging from 75 up to 90 dB). In contrast, no difference was observed when the sc-control group was compared to mice injected with anti-miR-99b together with *Zbp1* ASOs. Asterisks indicate significance (*P < 0.05). Error bars indicate SEM. Data information: Bars and error bars in panels (A, B) indicate mean ± SEM. Source data are available online for this figure.

anxiety. PPI is impaired in SZ patients and in mouse models for SZ (Light and Braff, 1999; van den Buuse, 2010). Furthermore, increased anxiety is a phenotype often observed in SZ patients (Achim et al, 2011), suggesting that reduced miR-99b-5p levels are indeed linked to the development of SZ-like phenotypes. Nevertheless, these data cannot establish a clear causal link between reduced miR-99b-5p levels and the pathogenesis of SZ in humans, since no animal model can fully recapitulate the complex processes in human patients due to functional and structural differences in the cortical anatomy (van Heukelum et al, 2020; Feifel and Shilling, 2010).

However, that miR-99b-5p is involved in the pathogenesis in SZ is further underscored by the results of our molecular analysis. RNA-seq analysis of the prefrontal cortex of mice revealed that inhibition of miR-99b-5p mainly led to an increased expression of genes, which is in agreement with the established action of miRs in controlling mRNA levels. Furthermore, the upregulated genes were almost exclusively related to immunity pathways in microglia, a process which has been linked to the pathogenesis of SZ. For example, immunohistochemical analysis revealed altered microglia in postmortem brain samples of SZ patients (Bayer et al, 1999). In addition, epidemiological data demonstrated a correlation between immune diseases and SZ (Benros et al, 2011), while several neuroimaging studies reported an increase in activated microglia in the brains of SZ patients (van Berckel et al, 2008; Ottoy et al, 2018). Finally, studies in animal models have implicated aberrant

microglia activation with the onset of SZ-like phenotypes (Juckel et al, 2011; Shelton et al, 2021). While miR-99b-5p has not been studied in microglia so far, these data are in line with previous reports demonstrating a role of the miR-99b in the modulation of inflammatory responses. For example, miR-99b levels are decreased in tumor-associated macrophages and re-expression of miR-99b attenuates tumor growth (Wang et al, 2020). Furthermore, inhibition of miR-99b in dendritic cells significantly elevated the levels of pro-inflammatory cytokines including *Il1ß* and *Tnfα* (Singh et al, 2013). These findings are in agreement with our data showing that inhibition of miR-99b-5p in the prefrontal cortex of mice increased the expression of pro-inflammatory cytokines including *Il1ß* and *Tnfα* in microglia that we had isolated from the brains of these mice via FACS. A strong upregulation of genes linked to inflammatory processes, including the upregulation of *Il1ß* and *Tnfα*, was also observed when miR-99b-5p was inhibited in IMG cells or primary microglia. This is interesting since increased *Ill1 ß* and *Tnfα* levels have been repeatedly reported in SZ patients (Momtazmanesh et al, 2019). For example, inhibition of TNFα was recently shown to ameliorate disease phenotypes in different mouse models of SZ (Shelton et al, 2021).

Aberrant microglia activation can affect neuronal function via synaptic pruning, a process that is based on the phagocytic activity of microglia (Vilalta and Brown, 2018). We observed that inhibition of miR-99b-5p in IMG cells and in primary microglia increased their phagocytic activity. Moreover, cortical neurons co-cultured with microglia that were treated with anti-miR-99b oligonucleotides displayed differentially expressed genes, of which the downregulated genes were linked to GO terms such as synapse assembly, regulation of synaptic plasticity or dendritic spine organization. As for the upregulated genes, the most significant GO term was synapse pruning. Since our data also revealed that neurons co-cultured with anti-miR-99b-treated microglia indeed displayed a reduced number of dendritic spines, our findings suggest a scenario in which reduced levels of miR-99b-5p lead to an upregulation of pro-inflammatory processes in microglia, which eventually impacts on synaptic structure. This interpretation is in agreement with previous reports suggesting that aberrant microglia activation leads to pathological synaptic pruning, which in turn leads to plasticity defects which could drive the pathogenesis of SZ (Sellgren et al, 2019; Inta et al, 2017). Notably, the increased expression of several complement factors in microglia have been implicated in this process (Germann et al, 2021). In line with these data, we observed increased expression of key complement factors in primary microglia and in corresponding microglia/neuron co-cultures in which miR-99b-5p was inhibited, as well as in postmortem brain samples from SZ patients. In summary, these findings provide a plausible mechanism on how reduced levels of miR-99b-5p can contribute to the pathogenesis of SZ, namely the induction of a pro-inflammatory response associated with synaptic pruning. Nevertheless, we cannot exclude that additional mechanisms within microglia or other neural cells play a role.

MiRs mediate their biological action by controlling the expression of specific target mRNAs. Our data showed that within microglia, miR-99b-5p controls the expression of the *Zbp1* gene that plays an important role in the innate immune response (Kuriakose and Kanneganti, 2018). ZBP1 acts as sensor for Z-DNA/Z-RNA and controls inflammatory pathways such as type I interferon-signaling and other pathways, eventually leading to the

upregulation of various pro-inflammatory cytokines including e.g., the induction of *IL1ß* (Takaoka et al, 2007; Kuriakose et al, 2016; Muendlein et al, 2021).

These data suggest that reduced levels of miR-99b-5p in microglia contribute to SZ-like phenotypes because the tight control of *Zbp1* levels is lost. In line with this interpretation, we demonstrated that the administration of *Zbp1*-ASO rescues the effects of anti-miR-99b treatment on SZ-like phenotypes in mice as well as in the corresponding cellular alterations observed in primary microglia from mice as well as in microglia derived from human iPSCs. Interestingly, the cellular processes we find to be affected by altered miR-99b-5p and *Zbp1* levels have also been implicated in other brain diseases. Thus, it will be important to investigate the role of miR-99b-5p and *Zbp1* in other neuropsychiatric diseases. Moreover, aberrant microglia activation and synaptic pruning is observed in neurodegenerative diseases such as Alzheimer's disease (Hong et al, 2016), and ZBP1 also controls the NRLP3 inflammasome (Kuriakose et al, 2016), a key regulator of neuroinflammatory phenotypes in Alzheimer's disease (Heneka et al, 2013). In this context it is interesting to note that one study found decreased miR-99b-5p levels in plasma samples obtained in a mouse model of Alzheimer's disease when measured at 6 and 9 months of age, while increased levels were reported in older mice (Ye et al, 2015). These data might underscore the need for further study as to the role of *miR-99b-5p* and *Zbp1* in microglia obtained from wild-type mice as well as mouse models for neuropsychiatric or neurodegenerative diseases at different ages. Indeed, it is well established that microglia undergo age-dependent functional changes and even differ between brain regions (Hart et al, 2012; Ayata et al, 2018).

There are other questions we could not address within the scope of this manuscript. It will for example be interesting to investigate the other candidate miRs we found in addition to miR-99b-5p. Similarly, it will be important to study the potential miR-99b-5p targets we found in addition to *Zbp-1* in the context of SZ. Another question relates to the mechanisms that underlie the downregulation of miR-99b-5p in SZ patients. In future projects it will be interesting to test for example whether miR-99b-5p is altered in SZ mouse models that are based on either genetic or environmental risk factors such as early life stress. In addition, it will be important to identify the source of elevated miR-99b-5p levels in blood samples of SZ patients. It is known that miRs can be transported from the brain to the periphery within exosomes (Mustapic et al, 2017; Bayraktar et al, 2017), and recent studies reported the isolation of microglia-derived exosomes from human blood (Kumar et al, 2021). While this approach is not undisputed, it will be interesting to apply such methods to the PsyCourse Study, which is, however, beyond the scope of the current work. Although our findings that miR-99b-5p is decreased in the brain and the blood of SZ patients support the idea that the changes in blood may reflect corresponding changes in the brain, we cannot conclusively answer this question at present. Rather, we suggest that the analysis of miR-99b-5p levels in blood may eventually help stratify patients for treatment, including novel approaches based on RNA therapeutics towards miR-99b-5p or *Zbp1*. Finally, it will be important to study whether miR-99b-5p plays a role in other neuropsychiatric diseases or if it is specifically deregulated in schizophrenia patients. Along the same line, we have to acknowledge that in our in vivo experiments the knock down

miR-99b-5p was not specific to microglia cells. While *Zbp1* expression in the brain is rather specific to microglia, miR-99b-5p expression is also observed in other cells. Thus, it will be interesting to study the role of miR-99b-5p in cell types other than microglia.

In conclusion, in the present study we identify a miR-99b-5p-*Zbp1* pathway in microglia as a novel mechanism that likely contributes to the pathogenesis of schizophrenia. Our data also suggest that strategies to increase the levels of miR-99b-5p or inhibit *Zbp1*, for example via ASOs, could serve as novel therapeutic strategies for treating SZ patients.

# Methods

## Human subjects

Data collection and analysis of the PsyCourse cohort were approved by the Ethics committees of the University Medical Center Goettingen (UMG), responsible for the clinical centers of the UMG, Bad Zwischenahn, Eschwege, Asklepios Specialized Hospital Goettingen, Hildesheim, Lüneburg, Liebenburg, Osnabrück, Rotenburg, Tiefenbrunn, and Wilhemshaven (Az: 24/8/14), and the Medical Faculty of the Ludwig Maximilians University Munich (LMU), responsible for the clinical centers Munich and Augsburg (Az: 087-14). The small RNAome analysis was approved by the Ethics Committee of the University Medical Center Göttingen (AZ 22/5/18). Blood samples (PAXgene Blood RNA Tubes; PreAnalytix, Qiagen) and behavioral data (Dataset EV1) of control and schizophrenia patients (98.01% were of european origin) were obtained from participants of the PsyCourse Study (Budde et al, 2018). Psychiatric diagnoses were confirmed using the Diagnosis and Statistical Manual of Mental Disorders Fourth Edition (DSM-IV) criteria. Control subjects were screened for psychiatric disorders using parts of the structured clinical interviews for mental disorders across the lifespan (MINI-DIPS). All subjects were assessed for psychiatric symptoms through a battery of standard tests including the Positive and Negative Syndrome Scale (PANSS), the Global Assessment of Functioning Scale (GAF), and the Beck depression inventory (BDI-II).

## Postmortem human brain samples

Postmortem tissue samples (prefrontal cortex A9&24) from controls ($n = 17$; 5 females & 12 males; age = $62.3 \pm 18.9$ years, PMD = $19.7 \pm 6.7$ h) and schizophrenia patients ($n = 13$; 5 females & 8 males; age $57.7 \pm 16.8$ years, PMD = $21 \pm 6.4$ h) were obtained with ethical approval and upon informed consent from the Harvard Brain Tissue Resource Center (Boston, USA). RNA was isolated using Trizol as described in the manufacturer protocol using the Directzol RNA isolation kit (Zymo Research, Germany). RNA concentration was determined by UV measurement. RNA integrity for library preparation was assessed using an RNA 6000 NanoChip in a 2100 Bioanalyzer (Agilent Technologies).

## High throughput small RNAome sequencing

Small RNAome libraries were prepared with total RNA according to the manufacturer's protocol with NEBNext® small RNA library

preparation kit. All human subject small RNAome libraries were prepared with 150 ng of total RNA. Briefly, total RNA was used as starting material, and the first strand of cDNA was generated, followed by PCR amplification. Libraries were pooled and PAGE was run for size selection. For small RNAome, ~150 bp band was cut and used for library purification and quantification. A final library concentration of 2 nM was applied for sequencing. The Illumina HiSeq 2000 platform was used for sequencing and was performed using a 50-bp single-read setup. Illumina's conversion software bcl2fastq (v2.20.2) was used for adapter trimming and converting the base calls in the per-cycle BCL files to the per-read FASTQ format from raw images. Demultiplexing was carried out using Illumina CASAVA 1.8. Sequencing adapters were removed using cutadapt-1.8.1. Sequence data quality was evaluated using FastQC (http://www.bioinformatics.babraham.ac.uk/projects/fastqc/). Sequencing quality was determined by the total number of reads, the percentage of GC content, the N content per base, sequence length distribution, duplication levels, overrepresented sequences and Kmer content.

## Data processing, QC, and differential expression (DE) analysis

Sequencing data was processed using a customized in-house software pipeline. Quality control of raw sequencing data was performed by using FastQC (v0.11.5). The quality of miRNAs reads was evaluated by mirtrace (v1.0.1). Reads counts were generated using TEsmall (v0.4.0) which uses bowtie (v1.1.2) for mapping. Reads were aligned to the Homo_sapiens GRCh38.p10 genome assembly (hg38). The miRNA reads were annotated using miRBase. Read counts were normalized with the DESeq2 (v1.26.0) package. Unwanted variance such as batch effects, library preparation effects, or technical variance was removed using RUVSeq for all data (v1.20.0; k = 1 was used for factors of unwanted variation). DeSeq2 was utilized for differential expression analysis and adjustment of confounding factors including age and sex. In the DESeq2 model, the PsyCourse data were corrected for sex, age, and medication in DeSeq2. Volcano plots were plotted with the R package EnhancedVolcano (v1.4.0).

## WGCNA analysis

microRNAome co-expression module analysis was carried out using the weighted gene co-expression network analysis (WGCNA) package (version 1.61) in R (Langfelder and Horvath, 2008). We first regressed out age, gender, and other latent factors from the sequencing data, and after that, normalized counts were log (base 2) transformed. Next, the transformed data were used to calculate pairwise Pearson's correlations between microRNAs and define a co-expression similarity matrix, which was further transformed into an adjacency matrix. Next, a soft thresholding power of 8 was chosen based on approximate scale-free topology and used to calculate pairwise topological overlap between microRNAs in order to construct a signed microRNA network. Modules of co-expressed microRNAs with a minimum module size of 10 were later identified using cutreeDynamic function with the following parameters: method = "hybrid", deepSplit = 4, pamRespectsDendro = F, pamStage = T. Closely related modules were merged using a dissimilarity correlation threshold of 0.25. Different modules were

summarized as a network of modular eigengenes (MEs), which were then correlated with the different psychiatric symptoms and functionality variables (e.g., PANSS, GAF, etc). The eigengene refers to the first principal component of the expression level of all the genes within a module. It therefore condenses the information from multiple transcripts into a single representative expression profile. The module membership (MM) of microRNAs was defined as the correlation of microRNA expression profile with MEs, and a correlation coefficient cutoff of 0.5 was set to select the module-specific microRNAs. The Pearson correlation of MEs and psychiatric symptoms and functionality variable was plotted as a heat map.

## Enriched gene ontology and pathways analysis

To construct the Gene Regulatory network (GRN) for miRNA-target genes we retrieved validated microRNA targets from miRTarBase (v 7.0) (http://mirtarbase.mbc.nctu.edu.tw/). micro-RNA target genes were further filtered based on the expression in the brain. Brain-enriched expression was examined using the Genotype-Tissue Expression (GTEx) database. (GTEx Consortium). To identify the biological processes and their pathways in the miRNA-target genes, the ClueGO v2.2.5 plugin of Cytoscape 3.2.1 was used (Shannon et al, 2003). In the ClueGo plugin (Bindea et al, 2009) a two-sided hypergeometric test was used to calculate the importance of each term and the Benjamini-Hochberg procedure was applied for the P value correction. KEGG (https://www.genome.jp/kegg/) and Reactome (https://reactome.org/) databases were used for the pathway analysis. To construct GRN for significantly deregulated mRNAs, the ClueGO v2.2.5 plugin of Cytoscape 3.2.1 was used. Biological processes (BP) and pathways with adjusted p value < 0.05 were selected for further analysis. For further analysis, cellular metabolism and cancer-related biological processes were omitted. Key BPs with low levels of GOLevel (because terms at lower levels are more specific and terms higher up are more general) were further considered for data presentation and interpretation.

## microRNA and mRNAs lipid nanoparticles preparation

LNA-based miR99b-5p inhibitor sequences (Qiagen) were used to inhibit the function of miR99b-5p. To decrease the expression of *Zbp1*, corresponding anti-sense oligos (ASO) were employed. ASOs, inhibitor, and negative control sequences were purchased from Qiagen. Sequences and catalog numbers can be found in Datasets EV16, 17). MicroRNA inhibitor, or ASOs lipid nanoparticle (LNP) formulation, was achieved using a proprietary mixture of lipids containing an ionizable cationic lipid, supplied as Neuro9™ siRNA Spark™ Kit (5 nmol). The miRNA inhibitor or ASOs were encapsulated using a microfluidic system for controlled mixing conditions on the NanoAssemblr™ Spark™ system (Precision Nanosystems, Canada). The experiments were performed as described in the manufacturer's protocol. Briefly, 5 nmol lyophilized microRNA inhibitor or ASOs were dissolved in formulation buffer 1 (FB1) to a final concentration of 2 nmol. This solution was further diluted to a final concentration of 930 μg/mL. Formulation buffer 2 (FB 2), microRNA inhibitor/ASOs in FB1, and lipid nanoparticles were added to the cartridge and encapsulated using the NanoAssembler Spark system.

## Animals

C57BL/6J mice were purchased from Janvier and housed in an animal facility with a 12-h light–dark cycle at constant temperature (23 °C) with *ad libitum* access to food and water. Animal experiments complied with relevant ethical regulations and were performed as approved by the Tierschutzbuero of the University Medical Center Göttingen in agreement with the Lower Saxony Ministry of Food and Agriculture (AZ 17/2733). All experiments were performed with 3 months-old male mice. Prefrontal cortex (PFC) region was dissected on day five after stereotaxic surgery for RNA-seq-based experiments.

## Stereotaxic surgery

For intracerebral stereotaxic injections of LNPs in the PFC, 3-month-old mice were anesthetized with Rompun 5 mg/kg and Ketavet 100 mg/kg. After application of local anesthesia to the skull, two small holes were drilled into the skull. Mice then received a bilateral injection of LNPs of microRNA inhibitor/negative control or ASOs (dose: 0.15 μg/mL for microRNA inhibitor/negative control; dose: 0.3 μg/mL for ASO+ microRNA inhibitor mix). LNPs were injected with a rate of 0.3 μl/min per side. Only 0.9 μl of LNPs were injected per hemisphere (0.5 μl/min). After surgery, all mice were monitored until full recovery from the anesthesia and housed under standardized conditions.

## Behavioral phenotyping

The open field test was performed to evaluate locomotory and exploratory functions. Mice were placed individually in the center of an open arena (of 1 m length, 1 m width, and side walls 20 cm high). Locomotory activity was recorded for 5 min using the VideoMot2 tracking system (TSE Systems). The elevated plus maze test was used to evaluate basal anxiety. Mice were placed individually in the center of a plastic box consisting of two open and two walled closed arms (10 × 40 cm each, walls 40 cm high). Their behavior was recorded for 5 min using the VideoMot2 system. Time spent in open versus closed arms was measured to assay basal anxiety phenotype. Prepulse inhibition (PPI) was performed to test the acoustical startle response (ASR). ASR was completed in an enclosed sound-attenuated startle box from TSA Systems. In brief, mice were placed individually inside a cage attached with a piezoelectric transducer platform in a sound-attenuated startle cabinet. These sensory transducers converted the movement of the platform induced by a startle response into a voltage signal. Acoustic stimuli were executed through speakers inside the box. The mice were given 3 min to habituate at 65 dB background noise and their activity was recorded for 2 min as baseline. After the baseline activity recording, the mice were tested to six pulse-alone trials, at 120-dB startle stimuli intensity for a duration of 40 ms. PPI of startle activity was measured by conducting trials for pre-pulse at 120 dB for 40 ms or preceding non-startling prepulses of 70, 75, 80, 85, 90 dB.

## RNA isolation

### Humans

PAXgene Blood RNA Tubes (PreAnalytix/Qiagen) were stored at −80 °C. For RNA isolation, the tubes were thawed and incubated at room temperature overnight. RNA was extracted according to the manufacturer's protocol using PAXgene Blood RNA Kits (Qiagen). RNA concentrations were measured by UV measurement. RNA integrity for library preparation was determined by analyzing them on an RNA 6000 NanoChip using a 2100 Bioanalyzer (Agilent Technologies).

### Mice

The mice were sacrificed by cervical dislocation on day five after stereotaxic surgery. Unilateral PFC region was collected and immediately frozen in liquid nitrogen and later stored at −80 °C until RNA isolation. Total RNA was isolated using the trizol method as described by the manufacturer's protocol using the Directzol RNA isolation kit (Zymo Research, Germany). The RNA concentration was determined by UV measurement. RNA integrity for library preparation was assessed using a Bioanalyzer (Agilent Technologies).

## RNA sequencing

Total RNA was used for the library preparation using the TrueSeq RNA library prep kit v2 (Illumina, USA) according to the manufacturer's protocol. 500 ng RNA was used as starting material. The quality of the libraries was assessed using the Bioanalyzer (Agilent Technologies). Library concentration was measured by Qubit™ dsDNA HS Assay Kit (Thermo Fisher Scientific, USA). Multiplexed libraries were directly loaded onto a Hiseq2000 (Ilumina) with 50 bp single-read setup.

The sequencing data were processed using a customized in-house software pipeline. Illumina's conversion software bcl2fastq (v2.20.2) was employed for adapter trimming and converting the base calls in the per-cycle BCL files to the per-read FASTQ format from raw images. Quality control of raw sequencing data was carried out using FastQC (v0.11.5) (http://www.bioinformatics.babraham.ac.uk/projects/fastqc/). Reads were aligned using the STAR aligner (v2.5.2b) and read counts were generated using featureCounts (v1.5.1). The mouse genome version mm10 was utilized.

## Publicly available datasets

Various publicly available datasets were used in this study to explore cell type-specific expression of differentially expressed genes. Published single-cell data (McKenzie et al, 2018) were utilized to explore neuron-, astrocyte-, and microglia-specific expression of genes. Immunome-related genes were retrieved from the Immunome database. The Immune Response In Silico (IRIS) dataset was used to explore immunity-related genes (Ortutay and Vihinen, 2006; Abbas et al, 2005).

## Primary microglia cultures

Primary mouse microglia cell cultures were prepared as previously described for wild-type pups (Islam et al, 2021). In brief, newborn mice (P1 pups) were used to prepare mixed glia cultures. Cells were grown in DMEM (Thermo Fisher Scientific) with 10% FBS, 20% L929 conditioned medium and 100 U ml⁻¹ penicillin–streptomycin (Thermo Fisher Scientific). Microglia were collected 10-12 days after cultivation by shake off, counted and plated in DMEM supplemented with 10% FBS, 20% L929 conditioned medium and

100 U ml$^{-1}$ penicillin–streptomycin. The microglia were shaken off up to two times.

## Ex vivo isolation of microglia

PFC regions were dissected, mechanically dissociated, and digested for 15 min with liberase (0.4 U/mL; Roche) and DNAse I (120 U/mL; Roche) at 37 °C. Subsequently, the cell suspension was passed through a 70 µm cell strainer. Myelin debris was eliminated by the Percoll density gradient. Single-cell suspension was labeled by using anti-mouse CD45 BV 421 (Clone 30-F11, Biolegend) and CD11b FITC (Clone M1/70, Biolegend). Antibody-labeled CD45$^{low}$ CD11b$^+$ microglial cells were sorted using a FACSAria 4 L SORP cell sorter (Becton Dickinson) The purity of the sorted microglial cells was above 90%.

## Primary neuronal culture

Primary neuronal cultures were prepared from E17 pregnant mice of CD1 background (Janvier Labs, France). Briefly, mice were sacrificed and the brains of embryos were taken out, meninges removed, and the cortex dissected out. The cortexes were washed in 1× PBS (Pan Biotech, Germany). Single-cell suspensions were generated by incubating them with trypsin and DNase before careful disintegration. One hundred and thirty thousand cells per well were plated on poly-D-lysine-coated 24-well plates in Neurobasal medium (Thermo Fisher Scientific, Germany) supplemented with B-27 (Thermo Fisher Scientific, Germany). Primary cortical neurons were used for experiments at DIV10-12.

## Magnetic-activated cell sorting (MACS) based cell sorting

Cells were isolated from the cortex of 3-month-old male C57B/6J mice using the adult brain dissociation kit (cat. no. 130-107-677, Miltenyi) according to the manufacturer's protocol with minor modifications. Briefly, mice were sacrificed using pentobarbital and the brains were quickly removed. To remove major parts of the meninges, the brains were rolled over Whatman paper and then the cortices were dissected and placed into the enzyme mixes. The tissue was incubated at 37 °C for 30 min in a water bath and triturated gently three times during this period. Then, the samples were applied to 40 µm cell strainers and the protocol was followed for debris and red blood cell removal. Oligodendrocytes were isolated using Anti-O4 microbeads (1:40, cat. no. 130-094-543), astrocytes using Anti-ACSA2 microbeads (1:10, cat. no. 130-097-678) and microglia with Anti-Cd11b microbeads (1:10, cat. no. 130-093-634). Purity of the cell type populations was determined by qPCR.

## Cell lines

All human iPSCs used in this study are commercially available and reported to be derived from material obtained under informed consent and appropriate ethical approvals.

## Differentiation of microglia from induced pluripotent stem cells

Human induced pluripotent stem cells lines (hiPSCs) (Cell line IDs: KOLF2.1 J (Pantazis et al, 2022) were obtained from The Jackson

Laboratory; BIONi010-C and BIONi037-A were both from the European bank for Induced Pluripotent Stem Cells) were differentiated to microglia as previously described (Haenseler et al, 2017). In brief, $3 × 10^6$ iPSCs were seeded into an Aggrewell 800 well (STEMCELL Technologies) to form embryoid bodies (EBs), in mTeSR1 and fed daily with medium plus 50 ng/ml BMP4 (Miltenyi Biotec), 50 ng/ml VEGF (Miltenyi Biotec), and 20 ng/ml SCF (R&D Systems). Four-day EBs were then differentiated in 6-well plates (15 EBs/well) in X-VIVO15 (Lonza) supplemented with 100 ng/ml M-CSF (Miltenyi Biotec), 25 ng/ml IL-3 (Miltenyi Biotec), 2 mM Glutamax (Invitrogen Life Technologies), and 0.055 mM beta-mercaptoethanol (Thermo Fisher Scientific), with fresh medium added weekly. Microglial precursors emerging in the supernatant after ~1 month were collected and isolated through a 40 µm cell strainer and plated in N2B27 media supplemented with 100 ng/ml M-CSF, 25 ng/ml interleukin 34 (IL-34) for differentiation.

## Quantitative PCR experiment

cDNA synthesis was performed using the miScript II RT Kit (Qiagen, Germany) according to the manufacturer's protocol. In brief, 200 ng total RNA was used for cDNA preparation. HiFlex Buffer was used so that the cDNA could be used for both mRNA and microRNA quantitative PCR (qPCR). A microRNA-specific forward primer and a universal reverse primer were used for quantification. The U6 small nuclear RNA gene was employed as an internal control. For mRNA quantification, gene-specific forward and reverse primers were used. The relative amounts of mRNA were normalized against GAPDH. The fold change for each microRNA and mRNA was calculated using the 2−ΔΔCt method. The Light Cycler® 480 Real-Time PCR System (Roche, Germany) was used to perform qPCR.

## Caspase 1 activation assay

Caspase-Glo® 1 Inflammasome Assay (Promega, Germany) was used to detect caspase 1 activation as described in the manufacturer's protocol. In brief, microglia, treated with ASO/inhibitor or primed with LPS and stimulated with ATP, were seeded on opaque, flat-bottom 96-well plates (Cellstar, Germany) at 50,000 per well in 100 µl DMEM supplemented with 10% FBS, 20% L929 conditioned medium and 100 U ml$^{-1}$ penicillin–streptomycin. 100 µl of Caspase-Glo buffer was mixed with cell medium. Plates were incubated at room temperature for 1 h. Luminogenic caspase activity was measured using a FLUOstar Omega plate reader (BMG Labtech).

## Microglia phagocytosis assay

The microglia phagocytosis assay was performed as described. Primary microglia cultures were plated at a density of $18 × 10^4$ in poly-D-lysine-coated 24-well plates in DMEM supplemented with 10% FBS, 20% L929 conditioned medium and 100 U ml$^{-1}$ penicillin–streptomycin. Immortalized microglia (IMG) cultures were plated at a density $5 × 10^3$ in poly-D-lysine-coated 24-well plates in DMEM supplemented with 10% FBS, 1X Glutamine (Millipore), and 100 U ml$^{-1}$ penicillin–streptomycin. To evaluate phagocytosis, treated microglia were incubated with fluorescent

latex beads of 1 µm diameter (green, fluorescent 496/519; Sigma-Aldrich) for 1 h at 37 °C, rinsed, and fixed with 4% formaldehyde. Cells were stained using the Iba1 (CD68) antibody (1:500; Wako) and DAPI. A confocal microscope was used for imaging at a low magnification (10×). ImageJ was used to quantify fluorescent latex beads. Region of interests (ROIs) were selected as microglial cells outlined with the Iba1 immunostaining to quantify beads. An intracellular section of the cell was selected to assure engulfment of latex beads by microglia. Similar acquisition parameters were used for each individual experiment. The results were expressed as the percentage of phagocytic index (# of total engulfed beads in an image/# of total cells identified in an image; n = 13 independent experiments).

## Synaptic pruning in primary microglia neural co-culture

Primary cortical neurons were seeded at a density of 130,000 on poly-D-lysine-coated 13 mm coverslips in 24-well plates in Neurobasal medium supplemented with B-27. Primary cortical neurons were used for experiments at DIV10-12. Treated primary microglia cultures were harvested from T-75 flasks and 4000 cells were seeded to each neural culture well. Plates were kept at 37 °C for 3 days. On the third day, the cells were washed and fixed with 4% PFA (Sigma-Aldrich, Germany) and 100 mM NH4Cl (Merck, Germany) respectively, at room temperature for 30 min. Next, the cells were washed in permeabilization and blocking buffer (0.1% Triton-X [Merck, Germany] + 3% bovine serum albumin (BSA) [AppliChem GmbH, Germany]) on a shaker. The cells were then incubated with primary antibodies for 1 h at room temperature. The antibodies used included synaptophysin 1 (guinea pig, SySy), PSD-95 (rabbit, Cell Signaling,), and Iba1 (goat, Abcam). After incubation, the cells were washed in PBS and then incubated with a secondary antibody for 1 h at room temperature. As secondary antibodies, Cy3 (donkey, anti- guinea pig, Jackson Imm.), Abberrior STAR 635p (goat, anti-rabbit) were used. Mowiol (Merck, Germany) and DAPI were used as a mounting medium. Images were taken with a multicolor confocal STED microscope (Abberior Instruments GmbH, Göttingen, Germany). Analysis of colocalization of pre- and post-synaptic markers were performed using SynQuant plugins in Fiji (v 2.0.0).

## Dendritic spine analysis

As described above, primary cortical neurons and primary microglia were co-cultured and fixed with 4%PFA. Dendritic spines were labeled as described (Goldberg et al, 2021). In brief, the cells were aspirated and 2–3 crystals of Dil stain (Life Technologies-Molecular Probes) were added to each culture well and incubated on a shaker for 10 min at room temperature. Cells were washed with PBS until no crystals were visible and incubated overnight at room temperature. On the following day, the cells were washed and mounted with Mowiol. For high-magnification images, a multicolor confocal STED microscope with a 60× oil objective was used. Spine density and total spine length were measured by using ImageJ.

## Protein extraction of primary microglia

Primary microglia cell lysates were used to detect ZBP1 in RIPA fractions. Primary microglia were seeded in a 6-well plate at a

density of $1 \times 10^6$ in each well. Cells were collected in a RIPA buffer supplemented with 1 x protease inhibitor. Samples were kept on ice for 15 min and vortexed every 5 min and then centrifuged at 5000 rpm for 15 min at 4 °C before supernatants were transferred to a new tube and stored at –20 °C. The protein concentration was measured using a BCA assay.

## Immunoblot analysis

For standard immunoblot analysis, 20 µg of samples were mixed with 1× Laemmli buffer (Sigma, Germany), heated for 5 min at 95 °C and loaded onto 4–15% Mini-PROTEAN® TGX™ Precast Protein Gels (Bio-Rad, Germany). Proteins were transferred on nitrocellulose membranes and membranes were blocked with 5% BSA in PBS-Tween. Membranes were incubated with primary antibodies in 5% BSA in PBS-Tween. Fluorescent-tagged secondary antibodies (LI-COR) were used for visualization of proteins. Imaging was performed using a LI-COR ODYSSEY. HSP-70, GAPDH were used as a loading and run on the same gel.

## Treatment of microglia

Microglia activation by LPS was used as a positive control. For this, microglia cells were first primed with 100 ng/ml ultrapure LPS (E. coli 0111:B4, Invivogen) and then incubated at 37 °C. After this, 5 mM ATP were added to the culture and incubated for 30 min. Caspase 1 assay and phagocytosis assay were performed from these cultures. For immunoblot, cell lysate was prepared. For miR99b-5p-related analysis, microglia were either treated for two days with miR99b-5p inhibitor/negative control or ASOs in T-75 after first harvesting or after harvesting cells were seeded in a 24-well culture plate.

## Luciferase assay

Seed sequences of miR-99b-5p and pairing 3′UTR sequences of Zbp1 were generated with TargetScan. Cloned 3′UTR sequence of Zbp1 and scrambles UTR were purchased from Gene Copoeia (https://www.genecopoeia.com/product/mirna-target-clones/mirna-targets/). UTR was cloned downstream to firefly luciferase of pEZX-MT06 Dual-Luciferase miTarget™ vector. The pEZX-MT06-scrambled UTR, pEZX-MT06-Zbp1 3′UTR or the mutated construct (cat. number: CS-MmiT101834-MT06-02-GC, Gene Copoeia) and miR99b-5p mimic or negative control were co-transfected into HEK293-T cells cultured in 24-well plates using EndoFectin™ Max Transfection Reagents (Gene Copoeia) according to the manufacturer's protocol. 48 h after transfection, Firefly and Renilla luciferase activities were measured using a Luc-Pair™ Duo-Luciferase HS Assay Kit (for high sensitivity) (GeneCopoeia). Firefly luciferase activity and Renilla luciferase activity were normalized. The mean of luciferase activity and of Firefly/Renilla was considered for the analysis.

## Statistical analysis

Unless otherwise noted, statistical analysis was carried out with GraphPad Prism software version 8.0. Statistical measurement is shown as mean ± SD. Each n represents a biological sample. A two-tailed unpaired t-test was used to compare two groups. If more than

2 groups were compared, a One-way ANOVA test was applied to analyze the data followed by post hoc analysis. Enriched gene ontology and pathway analysis was performed using Fisher's exact test followed by a Benjamini-Hochberg correction.

## Data availability

The RNA-seq data are available via the GEO accession number GSE224142. The RNA-seq data as well as metadata for the human samples including the data from the PsyCourse cohort cannot be shared anonymously due to data protection rules but are available via direct request to the Data sharing committee established by the Ludwig Maximilians University Munich. For this please direct an E-mail to Psy.Course-Proposals@med.uni-muenchen.de.

## Peer review information

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

## Acknowledgements

This work was supported by the following grants to AF: The DFG (*Deutsche Forschungsgemeinschaft*) project FI981-18, priority program 1738 (FI981-15), SFB1286 the EPIFUS project (FI981-13) and GRK2824, the Federal Ministry of Education and Research (BMBF) via the ERA-NET Neuron project EPINEURODEVO, the European Union Joint Programme-Neurodegenerative Diseases (JPND) – EPI-3E, Germany's Excellence Strategy - EXC 2067/1 390729940. FS was supported by the GoBIO project miRassay (BMBF, 16LW0055). Urs Heilbronner is supported by the European Union's Horizon 2020 Research and Innovation Programme (PSY-PGx, grant agreement No. 945151) and the Deutsche Forschungsgemeinschaft (DFG, German Research Foundation, project number 514201724). LE is supported by the Studienstiftung des Deutschen Volkes and the International Max-Planck Research School for The Mechanisms of Mental Function and Dysfunction (IMPRS-MMFD). DKV is supported by grants from the Chan Zuckerberg Initiative Neurodegenerative's Challenge Network (2020-221779(5022) & 2021-235147). TS was supported by grants from the *Deutsche Forschungsgemeinschaft* (DFG; SCHU 1603/4-1, 5-1, 7-1), the German Ministry of Education and Research (BMBF; 01EE1404H), and the Dr. Lisa Oehler Foundation (Kassel, Germany). PF was supported by a grant from the DFG (DFG; FA 241/16-1). SS was supported by the IMPRS-Genome Science (GS) program and RP was supported by the IMPRS-Neuroscience program.

## Author contributions

**Lalit Kaurani**: Conceptualization; Formal analysis; Investigation; Writing—original draft; Project administration; LK designed the study, conducted cell culture and mouse-related experiments, generated and analyzed sequencing data from human and mouse samples, produced and analyzed imaging data, conducted qPCR experiments, interpreted results, prepared figures, and drafted and revised the manuscript. **Md Rezaul Islam**: Data curation; Formal analysis; MRI performed stereotactic injections, carried out mouse-related experiments, analyzed WGCNA data, processed mouse PFC sequencing data, and analyzed imaging data. **Urs Heilbronner**: Data curation; Formal analysis; UH collected and evaluated phenotypic data within the PsyCourse cohort. **Dennis M Krüger**: Formal analysis; DMK curated PsyCourse sequencing data and assisted in its analysis. **Jiayin Zhou**: Formal analysis; Investigation; JZ assisted with primary neuronal and microglial cultures. **Aditi Methi**: Formal analysis; AM conducted WGCNA analysis. **Judith Strauss**: Formal analysis; Investigation; JS supported with primary microglial culture and FAC sorting of microglia. **Ranjit Pradhan**: Formal analysis; RP carried out Western blot analysis and performed revision work for Fig. EV3 and Fig. 4. **Sophie Schröder**: Formal analysis; Investigation; SS undertook MACS-based cell sorting and

qPCR for Appendix Fig. S4. **Susanne Burkhardt**: Investigation; SB helped with RNA sequencing experiments. **Anna-Lena Schuetz**: Investigation; ALS performed qPCR analysis. **Tonatiuh Pena Centeno**: Formal analysis; TP conducted and assisted with statistical analysis. **Lena Erlebach**: Investigation; LE cultured hiPSC-derived microglia. **Anika Bühler**: Investigation; AB cultured hiPSC-derived microglia. **Monika Budde**: Resources; Data curation; MB analyzed phenotypic data within the PsyCourse cohort. **Fanny Senner**: Resources; FS analyzed phenotypic data within the PsyCourse cohort. **Mojtaba Oraki Koshour**: Resources; MOK analyzed phenotypic data within the PsyCourse cohort. **Eva C Schulte**: Data curation; ECS analyzed phenotypic data within the PsyCourse cohort. **Max Schmauß**: Resources; MS analyzed phenotypic data within the PsyCourse cohort. **Eva Z Reinighaus**: Resources; Investigation; EZR analyzed phenotypic data within the PsyCourse cohort. **Georg Juckel**: Resources; GJ analyzed phenotypic data within the PsyCourse cohort. **Deborah Kronenberg-Versteeg**: Resources; Investigation; DKV cultured hiPSC-derived microglia. **Ivana Delalle**: Resources; ID provided postmortem brain samples from schizophrenia patients and control, contributing to the data presented in Fig. 1F. **Francesca Odoardi**: Investigation; FO conducted FAC sorting of microglia for Appendix Fig. S2. **Alexander Flügel**: Investigation; AF conducted FAC sorting of microglia for Appendix Fig. S2. **Thomas G Schulze**: Conceptualization; Resources; TGS secured funding, collected and analyzed phenotypic data within the PsyCourse cohort. **Peter Falkai**: Conceptualization; Resources; PF secured funding, collected and analyzed phenotypic data within the PsyCourse cohort. **Farahnaz Sananbenesi**: Conceptualization; Formal analysis; Writing—original draft; Project administration; FS secured funding, designed the study, supervised the research, interpreted results, and drafted and revised the manuscript. **André Fischer**: Conceptualization; Formal analysis; Supervision; Funding acquisition; Writing—original draft; Project administration; Writing—review and editing; AF secured funding, conceptualized and designed the study, provided supervision, interpreted results, prepared figures, and wrote and revised the manuscript.

## Funding

## Disclosure and competing interests statement
The authors declare no competing interests.

# Expanded View Figures

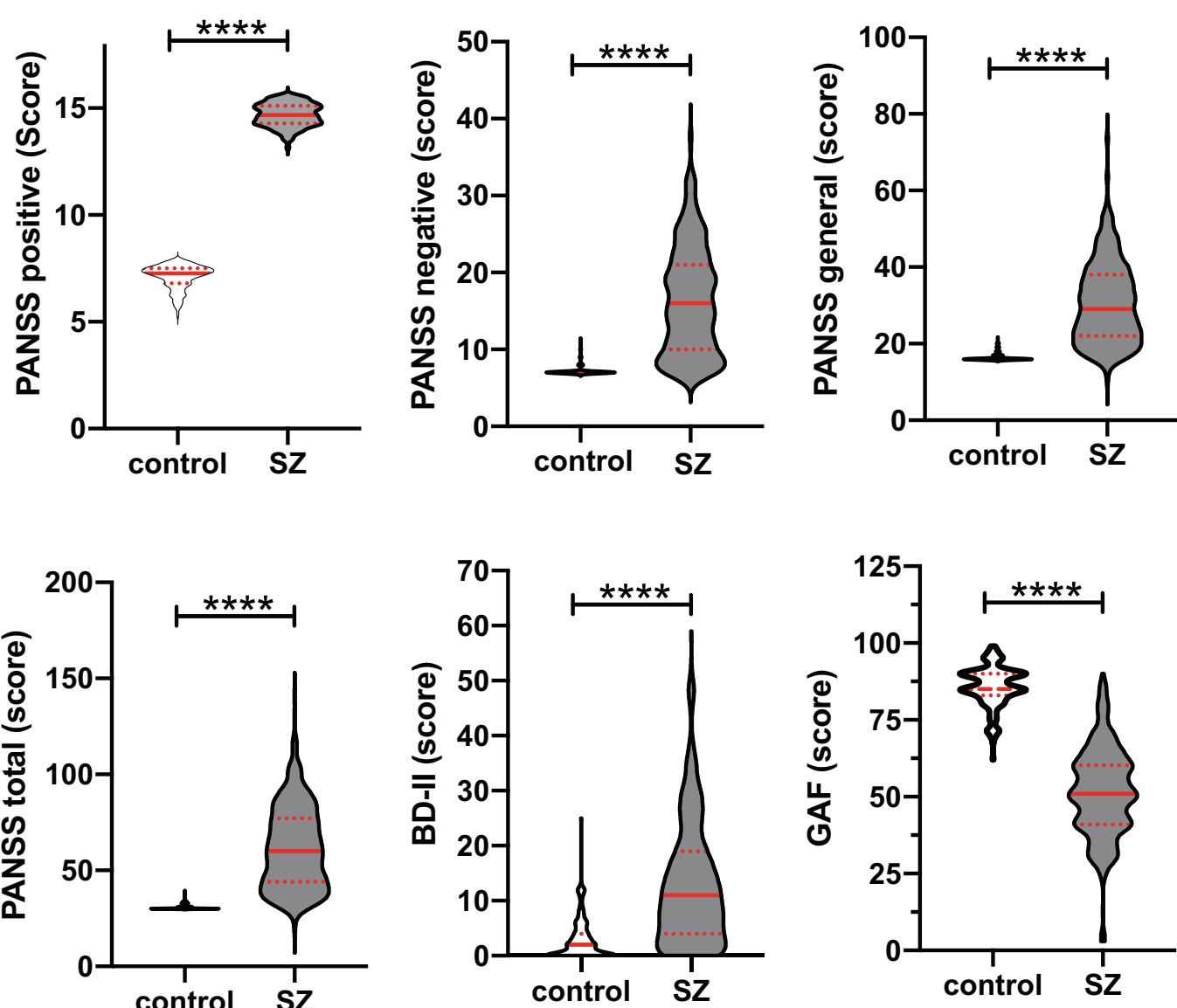

**Figure EV1.   Clinical phenotypes of the individuals subjected for small RNA-seq analysis.**

We analyzed 242 healthy controls and 331 SZ patients of the PsyCourse study. Depicted are the clinical phenotypes that differ significantly between groups, namely the positive and negative syndrome rating scale (PANSS), the total PANSS, the Beck depression inventory (BDI-II) and the global assessment of functioning (GAF) scores. ****$P < 0.0001$, *t*Test.

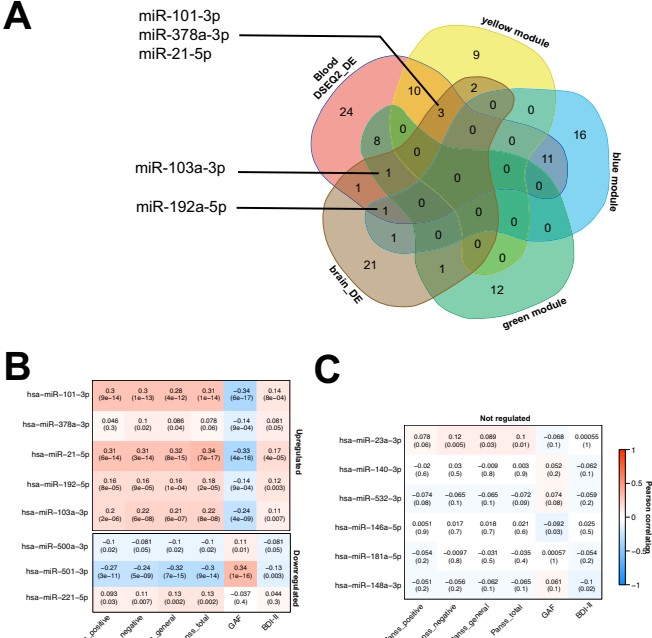

**Figure EV2. Analysis of candidate miR expression to SZ phenotypes.**

(A) Venn diagram showing the overlap between the miRs present within the co-expression modules that were increased in SZ patients (modules represented by yellow, blue and green color), that were increased when analyzed via DSEQ2 differential expression analysis in blood (Blood DSEQ2_DE) and were increased in postmortem brain samples from SZ patients (brain_DE). In total 5 miRs were consistently increased in blood, brain and a co-expression module (B) Heat map showing the correlation of candidate miR expression levels of individuals of the PsyCourse study to the clinical phenotypes. The numbers in each rectangle represent the correlation (upper number) and the corresponding p-value (lower number). Values for miR-99b-5p are shown within Fig. 1H. MiR-21-5p and miR-501-3p have been previously linked to SZ and show the highest correlation values. MiR-221-5p is significantly correlated to the PANSS scores but not the GAF and BDI-II (The significance of correlations was determined using Pearson correlation coefficients and assessed for statistical significance through permutation testing). (C) Heat map showing the correlation of six miRs from our dataset that were not differentially expressed in the blood or brain of SZ patients (not regulated). miR-23a-3p, miR-140-3p, and miR-532-3p were randomly selected, while miR-146a-5p, miR-181a-5p, and miR-148a-3p were recently identified as a biomarker signature for Alzheimer's disease (Islam et al, 2021) (The significance of correlations was determined using Pearson correlation coefficients and assessed for statistical significance through permutation testing).

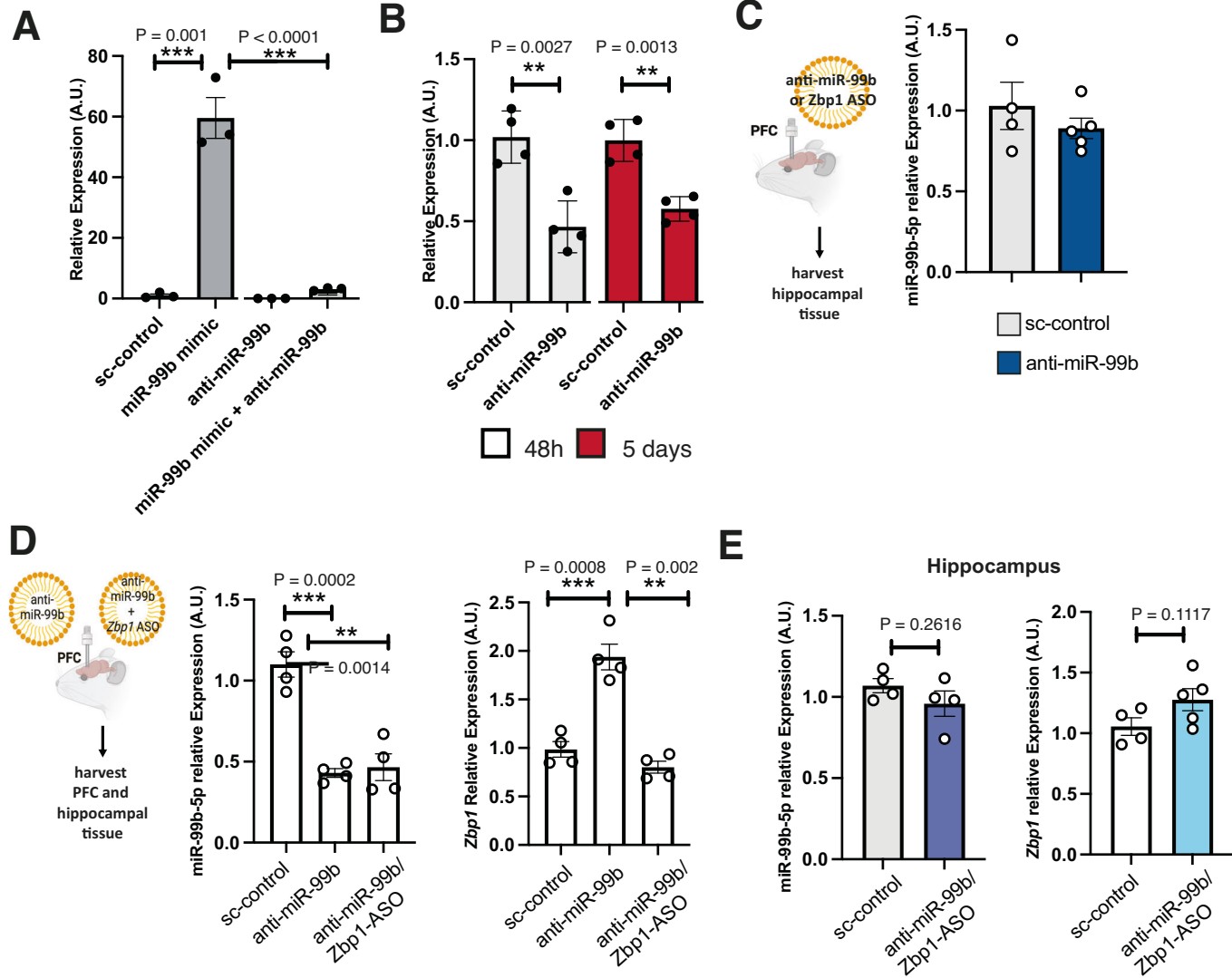

**Figure EV3.   Administration of anti-miR-99b-5p reduces the levels of miR-99b-5p detectable via qPCR.**

(A) To test if the administration of anti-miR-99b oligonucleotides reduces the levels of miR-99b-5p detected via qPCR we treated HEK293 cells—which show low endogenous expression of miR-99b-5p—with either control oligonucleotides (sc-control), miR-99b mimic, anti-miR-99b or the combination of miR-99b mimic and anti-miR99b. While administration of the miR-99b mimic significantly increased the detection of miR-99b-5p via qPCR when compared to sc-control ($P = 0.001$, t-Test), the co-administration significantly reduced detectable miR-99b-5p ($P < 0.0001$, t-test). $N = 3$/group. Measurements were performed 48 after treatment. (B) Primary mouse microglia (DIV 8) were treated with sc-control or anti-miR-99b for either 48 h or 5 days before RNA was collected for qPCR. At both time points anti-miR-99b treatment significantly reduced the detectability of miR-99b-5p ($n = 4$/group, t-test). It is important to note that LNA-based anti-miRs are known to sequester the target RNA rather than causing the degradation of the target miR. Therefore, the reduced detection of miR-99b-5p upon anti-miR-99b treatment is likely due to the formation of exonuclease-resistant duplexes that are not denatured during cDNA synthesis. Consequently, the miR-99b-5p is less accessible for reverse transcription, resulting in lower detected levels compared to the scramble control-treated samples in the subsequent qPCR reaction. Nevertheless, qPCR is a suitable method to assay target engagement of anti-miR-99b-5p. (C) Left panel: Experimental design. Anti-miR-99b oligonucleotides (anit-mIR-99b) were injected—along with their corresponding control—into the prefrontal cortex (PCF) of mice. Right panel: Bar graph showing qPCR results for miR-99b-5p in tissue obtained from the hippocampus of mice after injection of anti-miR-99b ($n = 4$) or sc-control oligonucleotides ($n = 5$) to the PFC. There is no significant difference between the groups. (D) Left panel: Experimental design. Anti-miR-99b or anti-miR-99b along with *Zbp1*-ASOs (anti-miR99b/*Zbp1*-ASO) were injected into the prefrontal cortex (PCF) of mice. Middle panel: QPCR results from the PFC show that the expression of miR-99b-5p is decreased in the anti-miR-99b group ($n = 4$; $P = 0.0002$, unpaired t-Test) and in the anti-miR99b/Zbp1-ASO group ($n = 4$; $P = 0.0014$, unpaired t-Test) when compared to the sc-control group ($n = 4$). One-way ANOVA revealed a significant difference among the groups ($p < 0.0001$). Right panel: QPCR results from the PFC show that the expression of *Zbp1* is increased in the anti-miR-99b group ($n = 4$; $P = 0.0008$, unpaired t-Test) and in the anti-miR99b/Zbp1-ASO group ($n = 4$; $P = 0.002$, unpaired t-Test) when compared to the sc-control group ($n = 4$). One-way ANOVA revealed a significant difference among the groups ($p < 0.0001$). (E) Right panel: Bar graph showing qPCR results for *miR-99b-5p* in tissue obtained from the hippocampus of mice after injection of anti-miR-99b along with *Zbp1* ASOs (anti-miR99b/Zbp1-ASO group; $n = 4$) or control oligonucleotides ($n = 4$) into the PFC. There is no significant difference between the groups. Left panel: Bar graph showing qPCR results for *Zbp1* in tissue obtained from the hippocampus of mice after injection of anti-miR-99b along with *Zbp1* ASOs (anti-miR99b/Zbp1-ASO group; $n = 4$) or control oligonucleotides ($n = 5$) into the PFC. There is no significant difference between the groups. Data information: Bars and error bars in panels (A–E) indicate mean ± SEM.

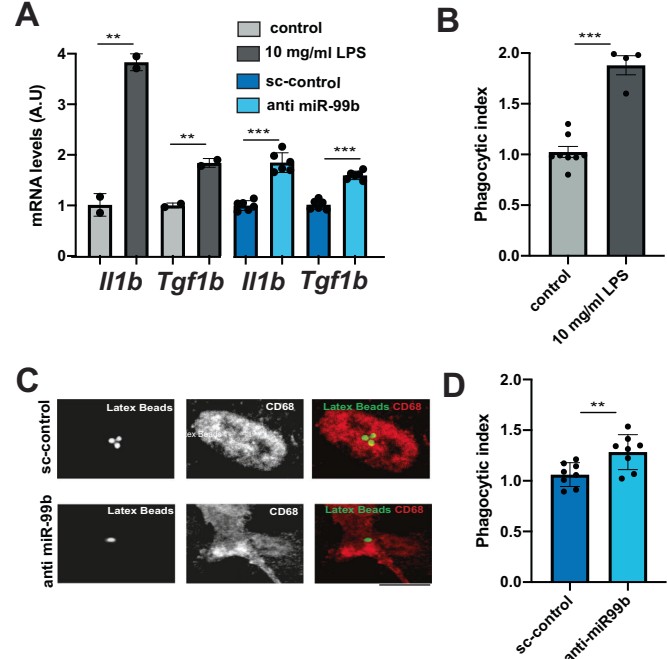

**Figure EV4.   Loss of miR-99b-5p levels increases the expression of pro-inflammatory cytokines and phagocytosis in IMG cells.**

(**A**) IMG cells were treated with vehicle control solution (control) or 10 mg/ml LPS. In a similar experiment, IMG cells were treated with LNPs loaded with either sc-control LNAs or anti-miR-99b. qPCR analysis was performed for the pro-inflammatory cytokines *Il1b* and *Tgf1b*. The expression of *Il1b* and *Tgf1b* were significantly increased upon LPS treatment or anti-miR-99b treatment when compared to the respective control groups (unpaired t test; $n = 2$ or 6/group). (**B**) Bar graph showing that the phagocytic index increases in IMG cells upon LPS treatment (unpaired t test; controls = 8; LPS = 4). (**C**) Representative images showing the uptake of latex beads by IMG cells treated with either sc-control LNAs or anti-miR-99b. Scale bar: 0.5 μm. (**D**) Bar graph showing the quantification of (**C**). The figure presents representative images captured at 63× magnification using a confocal microscope. A scale bar of 0.5 μm is included in each image to provide a reference for size. Treatment with anti-miR-99 increases the phagocytic index (unpaired t test; $n = 8$/group). *$P < 0.05$, **$P < 0.01$, ***$P < 0.001$. Error bars indicate SEM. Data information: Bars and error bars in panels (**A**, **B**, **D**) indicate mean ± SEM.

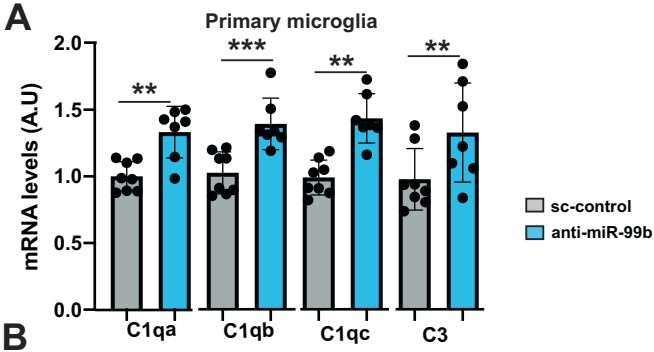

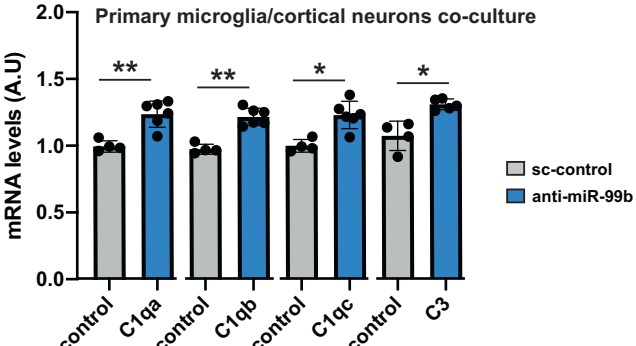

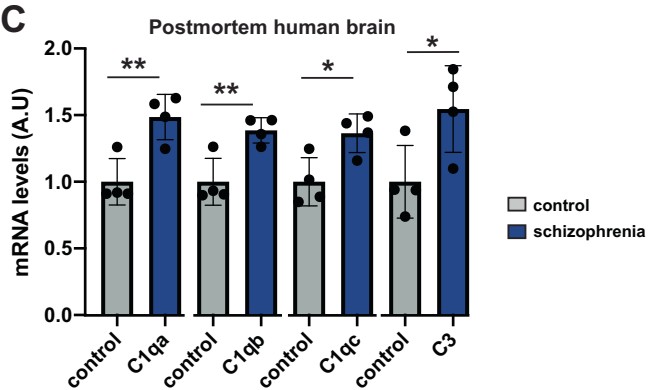

**Figure EV5. Genes linked to innate immunity and synaptic pruning are increased in microglia treated with anti-miR-99b and in schizophrenia patients.**

(A) Bar graph showing qPCR results for *C1qa*, *C1qb*, *C1qc*, and *C3* in primary microglia treated with sc-control LNAs or anti-miR-99b. (B) Bar graph showing qPCR results for *C1qa*, *C1qb*, *C1qc*, and *C3* when RNA was isolated from co-cultures in which primary cortical neurons were treated with microglia that had received LNPs loaded with either sc-control LNAs or anti-miR-99b. (C) QPCR analysis was used to measure *C1qa*, *C1qb*, *C1qc*, and *C3* expression in human postmortem PFC samples obtained from control individuals and SZ patients (unpaired *t*Test, **$P < 0.01$, *$P < 0.05$; $n = 4$/group). Data information: Bars and error bars in panels (A, B, C) indicate mean ± SEM.

