## [Peer Review File · The EMBO Journal]

Regulation of Zbp1 by miR-99b-5p in microglia controls the development of schizophrenia-like symptoms in mice

Lalit Kaurani, Md Rezaul Islam, Urs Heilbronner, Dennis Krüger, Jiayin Zhou, Aditi Methi, Judith Strauss, Ranjit Pradhan, Sophie Schröder, Susanne Burkhardt, Anna-Lena Schuetz, Tonatiuh Pena Centeno, Lena Erlebach, Anika Bühler, Monika Budde, Fanny Senner, Mojtaba Oraki Koshour, Eva Schulte, Max Schmauß, Eva Reinighaus, Georg Juckel, Deborah Kronenberg-Versteeg, Ivana Delalle, Francesca Odoardi, Alexander Flügel, Thomas Schulze, Peter Falkai, Farahnaz Sananbenesi, and André Fischer

Corresponding author(s): André Fischer (A.Fischer@eni-g.de) , Peter Falkai (Peter.Falkai@med.uni-muenchen.de), Farahnaz Sananbenesi (farahnaz.sananbenesi@dzne.de), Lalit Kaurani (lalit.kaurani@dzne.de), Thomas Schulze (Thomas.Schulze@med.uni-muenchen.de)

Review Timeline:

Submission Date:	3rd May 23
Editorial Decision:	2nd Jun 23
Revision Received:	19th Dec 23
Editorial Decision:	24th Jan 24
Revision Received:	9th Feb 24
Accepted:	20th Feb 24

Editors: Karin Dumstrei / Ioannis Papaioannou

Transaction Report:

Dear Andre,

Thank you for submitting your manuscript to The EMBO Journal. Your study has now been seen by three referees and their comments are provided below.

As you can see the referees find the analysis interesting and suitable for consideration. The referees raise a number of constructive comments that I would like to ask you to address in a revised version. Let me know if we need to discuss anything specific.

Thank you for the opportunity to consider your work for publication. I look forward to your revision.

with best wishes

Karin

Karin Dumstrei, PhD
Senior Editor
The EMBO Journal

We realize that it is difficult to revise to a specific deadline. In the interest of protecting the conceptual advance provided by the work, we recommend a revision within 3 months (31st Aug 2023). Please discuss the revision progress ahead of this time with the editor if you require more time to complete the revisions.

As a matter of policy, competing manuscripts published during this period will not negatively impact on our assessment of the conceptual advance presented by your study.

Use the link below to submit your revision:

Referee #1:

Kaurani et al suggest in "A novel miR-99b-5p-Zbp1 pathway in microglia contributes to the pathogenesis of schizophrenia" that miR-99b-5p is downregulated in both the prefrontal cortex and blood of patients suffering from schizophrenia. Accordingly, inhibition of miR-99b-5p leads to a schizophrenia-like phenotype in mice. miR-99b-5p mediate inflammatory response in microglia, upstream of Zbp1, and Zbp1 knockdown ameliorates the pathological phenotypes caused by miR-99b-5p inhibition. The discovery of a novel miR-99b-5p-Zbp1 axis in microglia that might be relevant to the pathogenesis of schizophrenia is novel and interesting.

Major concerns

Cohorts size:

Study of 331 healthy controls and 242 SZ patients is in good numbers and is satisfying. Autopsy materials of 13 SZ patients and 17 controls is satisfying.

Results of Figure 1: WGCNA study (lines 107-137) lacks scientific conclusions

Figure 2A: that miR-99b-5p levels were significantly decreased in the PFC when measured 5 or 10 days after the injections of anti-miR-99 LNA oligos (Fig 2a) is unexpected. While LNAs are known to inhibit miRNA activity, they should not drive degradation/ decay of mature miRNAs. How can this be explained? Authors should demonstrate the downregulation effect in a controlled manner, for example in cultured microglia with repeated longitudinal analysis of miR-99 levels.

Figure 2: Distance traveled, Time spent in center/ open arms, and pre-pulse inhibition of the startle response are all signs of anxiety. So there is no evidence from experiments in animals that connects miR-99 and SZ. Accordingly, the statement 'decreased expression of miR-99b-5p is causatively linked to the development of SZ-like phenotypes.' Is incorrect (line 170)

Figure 4: the authors provide is insufficient data to make the case that miR-99b-5p can directly regulate Zbp1 levels (line 278). Prediction algorithms including Targetscan, the most established target prediction algorithm, do not predict Zbp1 downstream of miR-99/100. Neither there is evidence in miRDB. . In fact based on the authors Fig 4b, where a G:U wobble is observed where the continuous seed should be it is more likely that Zbp1 is not a target of miR-99 and it would be a mistake to claim it is.

https://www.targetscan.org/cgi-bin/targetscan/vert_80/targetscan.cgi?species=Human&gid=&mir_sc=miR-99-5p%2F100-5p&mir_c=&mir_nc=&mir_vnc=&mirg=

[https://www.targetscan.org/cgi-bin/vert_80/view_genetable.cgi?
rs=ENST00000371173.3&taxid=9606&members=&subset=1&showcnc=1&shownc=1&sortText=cs](https://www.targetscan.org/cgi-bin/vert_80/view_genetable.cgi?rs=ENST00000371173.3&taxid=9606&members=&subset=1&showcnc=1&shownc=1&sortText=cs)
<https://mirdb.org/cgi-bin/search.cgi>

Authors should justify their choice and substantiate the evidence for true target:miRNA regulation

The authors should demonstrate an accepted, continuous, miR-99 binding site at the 3'UTR of Zbp1. Then mutate the seed binding site and demonstrate abrogation of regulation. In brain tissue they should provide evidence for direct engagement of Zbp1 with miRNA regulation at the same sequence. For example by data mining of published Ago2 CLIP studies. Other options include unambiguous identification of mirna:target site interactions by different types of ligation

Reactions <https://www.ncbi.nlm.nih.gov/pmc/articles/PMC4181535/>

In summary, reasonably well-supported conclusions: miR-99 affect Zbp1 levels and anxiety-like behavior in mice and in microglia. Not sufficiently supported are the hypotheses that miR-99 is specifically involved in SZ and that miR-99 is targeting Zbp1 directly.

Minor comments

Writing of the main text with terms that refer to graphics of the figure (lines 113-114 "The turquoise, yellow, blue and red" should be changed to meet standard practice. Line 128 "Taken together, these data suggest that especially the 128 microRNAs

present in turquoise, pink, green, blue, yellow and red modules warrant further analysis." This is completely obscure to the reader. Re-write with scientific-relevant conclusions

Referring to cell free miRNA as "expression" is wrong; as these might be spilled out of cells and any way do not obey "regulation of gene expression" terms. Simply use "levels".

Line 159-161 Expect for miR-500a-159 3p and miR-221-5p, the expression of all other candidate miRs have is significantly correlated to the 160 PANSSs, GAF and BDI-II scores" delete "have"

How "GO analysis of the downregulated genes did not yield any highly significant pathways but detected processes linked to voltage-gated potassium channels (line 194) is suggesting immune-related processes (line 196) should be better explained.

Reference 34 as link of microglia to SZ is weak (R. Rahimian, M. Wakid, L. A. O'Leary, and N. Mechawar, "The emerging tale of 996 microglia in psychiatric disorders.," *Neurosci Biobehav Rev.*, vol. 131, pp. 1-29, 2021.) and should be substantiated by better more central citations of literature.

After 3h the authors state that microglia lacking miR-99b-5p may have detrimental effects on synaptic function when co-cultured with cortical neurons (line 245). However, the needed evidence for that is only in Fig 3i so the conclusions should be positioned after 3i.

Rephrase broken sentence: GO term analysis was performed for the 13 genes and revealed that they are linked to inflammatory processes including type I interferon signaling (Fig. EV7, supplemental table 14), linked to schizophrenia [37] [38]. (lines 263,264).

Discussion is long (lines 316-457) and can benefit from editing and focusing

Referee #2:

Comments:

In this study, Kaurani et al. identify a miRNA -miR99b- that is down-regulated in Schizophrenia (SZ) patients and may play an important role in the pathogenesis of SZ by promoting pro-inflammatory responses via its target Zbp1. The work stems from the sequencing of the MicroRNAome from blood and brain samples of two different sets of SZ patients and then focuses on dissecting the molecular pathway downstream of miR99b and its relevance for SZ.

To this goal, the authors use a combination of in-vivo and in vitro approaches, spanning from mice behavioral experiments to bioinformatics analyses and tests of microglia activity. Overall, the manuscript is interesting and can count on two major strength points. Firstly, the data gathered from the patients provide a precious resource for further studies, and their analysis a useful guideline on how one could integrate different types of biological information. Secondly, the body of evidence collected by Kaurani and colleagues in favour of a role for miR99-b and Zbp1 in SZ, summed to the fact that this pathway has never been studied before in the context of SZ, makes the study of relevance to deepen our understanding of a disease still orphan of effective treatments such as SZ.

Nevertheless, the enthusiasm towards the work as a whole is dampened by the presence of some experimental issues which need to be addressed to meet the standards of EMBO J. In addition, we think that the presentation of the data, quality of figures and writing could also be improved. Below we detail our comments, hoping that this helps the authors to improve the manuscript.

Main points:

- The main claim of the paper - the microglia-mediated effects of miR-99b downregulation in SZ - is mostly grounded on in-vitro experiments which are overall well-designed and executed. Nevertheless, we think that some of the in-vitro results need to be substantiated by further in-vivo evidences. To us it is unclear, for example, why the authors did not investigate the presence of pro-inflammatory changes and/or morphological changes like spine density in the brains of mice injected with the anti miR-99b. Furthermore, to what extent miR-99b expression is cell-type specific is not evident from the data provided, and could be readily addressed by combining imaging technologies for detection of microRNAs with cell-type markers labelling. The same is true also for the downstream target Zbp1, for which no details are provided concerning its expression in the PFC, with the exception of the qPCR of figure EV8a.

- At line 117-118, the authors state that the clinical phenotypes PANSS and BDI-II are decreased in SZ patients and GAF is increased. However, from the graphs in figure EV1 it appears clearly the opposite: PANSS and BDI-II are increased in SZ patients and GAF is decreased. Moreover, on the graph plotting the PANSS score the group labels (SZ and control) are inverted. From the DE analysis of SZ patients the authors identify 93 (blood) and 68 (brain) differentially regulated miRNAs (fig 1e, 1f). These numbers are at odds with what reported in fig 1g, where a total of 34 miRNAs from blood and 9 from brain is reported. Can the authors clarify?

- Related to the point above, where is the demographic information for the human blood samples? Were samples age-, sex- and ethnicity-matched, for example?

- It appears that while for the behavior a LNA-based miR-99b-5p was used, an ASO-based inhibitor was used for the RNA

sequencing data. This is problematic on several fronts. First, the data generated this way cannot be directly compared. Second, what is the evidence that the two inhibiting systems perform equally well? At the very least, the authors have provide evidence for equipotency of inhibition of the two approaches, or better, repeat one of the experiments with the other inhibitor.

- Related to the above, for the anti-miR99b experiment of figure 2 and the Zbp1 ASO rescue experiment of figure 4, did the authors verify surgery efficiency -ie. check that only PFC is targeted, that this is comparable amongst the different mice groups? Moreover, did they check the levels of Zbp1 expression itself following the ASO manipulation for both the in-vitro and the in-vivo experiments?

- In the experiment of figure EV4, to which sample does the FACS plot refers to? Please provide plots for both experimental groups, as well as the negative control used for setting the gates (eg. Isotype control). Moreover, it is unclear how the microglia cell numbers in EV4 b were determined. We suggest an immunohistochemistry approach to achieve a more direct quantification.

- In some cases, the statistical tests chosen are not coherent with the way data are plotted. For example in figure 2a, where two independent variables are present (days and treatments) on the same graph. The data should be analyzed using a two-way ANOVA, and mir-99b expression values be expressed relative only to the 5 days control timepoint. Yet mir99b values at each timepoint are plotted relatively to each matched control (5 days and 10 days), and t-test is used. The same happens in figure 2j. Data are analyzed by t-test, but plotted on the same graph. In other cases, the statistical information is missing / not clearly stated. For example, is the difference in Zbp1 expression detected by WB following anti miR99b significant (figure 4c)? Also, from the legend is not clear if a t-test was used for data in figure 4f-4k. If that's the case, it would have not been more appropriate to perform a one-way anova, since 3 groups were compared? Finally, could the authors clarify how the PPI data from figure EV8 were analyzed? From the legend, it seems that they used a combination of one-way Anova, two way Anova and t-test. This approach seems unusual and it is at odds with the other PPI experiment shown in figure2, where only t-test was used.

Minor points:

- We find that the manuscript is at places of a difficult read. Many paragraphs are written as a concatenation of information, and not well structured. This should be improved as it will ultimately help the reader (and reviewer). Here are the most worrisome examples: A) spelling/grammar mistakes (eg. line 159, line 329) B) the use of expressions not precise enough for a scientific article (eg. line 41 "vast majority"; line 65 "most of the"; line 172 "representing"; what is the difference between "primary microglia" (line 217) and "primary microglia" (line 230)? The way it's written, it seems that these are two different experiments, while in reality they are not: line 364 "essentially nothing was known") C) Sentences juxtaposed using elusive logical implications (eg. the word "however" on line 41 is not the right connecting word, as the information presented in this sentence is not contradicting the information provided in the previous one. (On line 65, in contrast, "however" is used the right way, as both sentences are referring to the same type of information.); from line 80 to 89, from line 320 to 322, from 424 to 426.) D) A substantial part of the discussion section is dedicated to exhaustively reviewing the literature on the role of other miRNAs in SZ, when other highly relevant topics (eg. the mechanisms causing miR-99b and/or miRNAs deregulation in general in SZ, state of the art of miR-based therapeutic/diagnostic strategies in psychiatric disorders and/or possible approaches for rescuing miR-99b expression and/or block Zbp1 in SZ) are dealt with in few lines or not touched at all.

- For the sequencing experiment of figure 2g, were the animals pooled together? If yes, how many mice per group? Moreover, how many replicates of the RNA-seq data were used to do the DEG analysis? Please provide these details in the main text and/or figure legend.

- We find that the general layout of the figures as well as the level of information provided in the figures and/or figure legends could be improved to be more intuitive and easier to interpret. A few examples A) Figure 1. In 1b-c, for some clusters it is possible to see the individual data points, for others is not. Please make it consistent. Also, the abbreviation ME (= expression module) is not clearly spelt out in the legend, and the technical term "eigen-expression" is not explained either. This makes the first part of figure 1b not very intuitive. Perhaps the authors could include in the main figure 1 or in the supplementary a scheme for the bioinformatic analysis encompassing the results of 1b-1d. Finally, the graphs in 1e and 1f could benefit from a title and a legend. B) Figure 2. It is difficult to see the individual data points for 2b-2f and 2j. Please change colors for the anti miR-99b group. For 2g, could the authors indicate which dots correspond to Il1b, Tgfb1 and Tnfa, since they claim in the main text that these genes are differentially expressed. As for 1e and 1f, and 2g, they could be improved with title and legend. C) Figure EV6. Plotting all the qPCR data for different genes on the same graph is misleading. The authors should at least indicate for each pairwise comparison the gene measured, as done in figure EV6a. D) Figure 3i: Absolute values of dendritic spines must be given per given segment. It is not clear what the number of dendritic spines was normalized to, nor the statistical test used.

- Lastly, the discussion is too long and off target. For example, all the text pertaining to microRNAs other than miR-99b-5p can be cut.

Referee #3:

The study by Kaurani et al. identifies the miR-99b-5p-Zbp1 pathway as contributing to the pathogenesis of schizophrenia. Specifically, their results show that levels of miR-99b-5p are reduced in blood and postmortem brains from schizophrenia patients while pro-inflammatory genes (including Zbp1) are upregulated. To establish a causal link between miR-99b-5p and schizophrenia-related behaviors, they went on by inhibiting miR-99b-5p in the prefrontal cortex of 3-months old mice and found that this led to schizophrenia-related phenotypes and was associated with an increase in the expression of pro-inflammatory genes that are enriched in microglia. The authors could confirm those results in in vitro preparations and further demonstrated

that inhibiting miR-99b-5p in microglia leads to increased phagocytosis and decreased spine number and was again associated with an upregulation of genes associated with neuro-inflammatory processes and synaptic pruning. From their RNAseq data, the authors identified the pro-inflammatory gene Zbp1 as a direct target of miR-99b-5p. Importantly, they could demonstrate that silencing Zbp1 rescues the effects of inhibiting miR-99b-5p on caspase activity, IL1beta expression and phagocytosis in both primary or human iPSC-derived microglia. Importantly, they finally demonstrate that silencing Zbp1 also rescues behavioral deficits induced by inhibiting miR-99b-5p in the prefrontal cortex of mice.

Overall, this is an important, impressive and elegant study that combines molecular manipulations to high throughput transcriptomic and refined bioinformatics analysis in both in vivo and in vitro models. The experimental approach is elegant and compelling and the conclusions are well supported by the data. The fact that many of the results could be recapitulated in both human and mouse models, both in vivo and in vitro underscores the importance and robustness of the findings. Not only do the authors identify a new signaling pathway that likely contributes to schizophrenia pathogenesis but they also validate other miRNA candidates that were previously linked to schizophrenia and other brain disorders. I would recommend the paper for publication provided that the following minor concerns are addressed:

- P8-L274 - 'Co-transfection of this vector with miR-99b-5p LNAs significantly reduced the luciferase activity (Fig 4b), but this was not the case when scramble control LNAs were used'. I believe that in the luciferase assay, the luciferase reporter construct carrying the Zbp1 UTR or a scramble (control) UTR was co-expressed with a miR-99b-5p mimic, not the miR-99b-5p LNA. This should be corrected.
- Fig. 3E - The authors should illustrate the phagocytosis assay with representative images.
- Fig. 3i - It would be nice to test whether Zbp1-ASO could also rescue the decrease in spine density to also have a cellular readout of the regulation of Zbp1 by miR-99b-5p at the neuronal level.
- The strategies used by the authors to inhibit miR-99b or Zbp1 is not specifically targeted to microglial cells. The authors should thus discuss in more details the relative expression of miR-99-5p in microglia versus neurons or other glial cell types and discuss their results in light of a potential action of anti-miR99b or Zbp1-ASO in cells other than microglia.
- I find the data shown in Fig. EV8 b,c important as it demonstrates the role of the regulation of Zbp1 by miR-99b-5p in schizophrenia-related behaviors. I would recommend to show them in a main figure.

Point to point rebuttal**Reviewer 1**

Kaurani et al suggest in "A novel miR-99b-5p-Zbp1 pathway in microglia contributes to the pathogenesis of schizophrenia" that miR-99b-5p is downregulated in both the prefrontal cortex and blood of patients suffering from schizophrenia. Accordingly, inhibition of miR-99b-5p leads to a schizophrenia-like phenotype in mice. miR-99b-5p mediate inflammatory response in microglia, upstream of Zbp1, and Zbp1 knockdown ameliorates the pathological phenotypes caused by miR-99b-5p inhibition.

The discovery of a novel miR-99b-5p-Zbp1 axis in microglia that might be relevant to the pathogenesis of schizophrenia is novel and interesting.

Major concerns

Reviewer 1 comments on the cohort size.

"Cohorts size:

Study of 331 healthy controls and 242 SZ patients is in good numbers and is satisfying. Autopsy materials of 13 SZ patients and 17 controls is satisfying."

We agree with this reviewer.

Reviewer 1 comments on Figure 1 and says:

"Results of Figure 1: WGCNA study (lines 107-137) lacks scientific conclusions"

The WGCNA analysis along with the subsequent differential expression analysis and the comparison of data from blood and postmortem human brain tissue represent the exploratory part of our study. However, we understand from this comment that the description of these data should be improved for clarity with a specific focus on the aims and the conclusion.

Therefore we have now substantially revised this part and explicitly highlighted the aims as well as the scientific conclusions of our analysis.

Please see page 4 lines 9-34 and page 5 lines 1-7 of the revised manuscript. Changes are underlined.

Reviewer 1 comments on Figure 2A and says:

"Figure 2A: that miR-99b-5p levels were significantly decreased in the PFC when measured 5 or 10 days after the injections of anti-miR-99 LNA oligos (Fig 2a) is unexpected. While LNAs are known to inhibit miRNA activity, they should not drive degradation/ decay of mature miRNAs. How can this be explained? Authors should show the downregulation effect in a controlled manner, for example in cultured microglia with repeated longitudinal analysis of miR-99 levels."

We thank this reviewer for bringing up this insightful observation and we agree that our description of these data was not sufficient.

While oligonucleotides that complementary bind to miRs, such as LNA-based oligos, are generally understood to inhibit miRNA function without the necessity to induce their degradation, there are plausible explanations for the observed decrease in miR-99b-5p levels when measured via qPCR.

To this end, we'd like to bring attention to an important aspect of the LNA methodology. The LNA-based anti-miR-99b sequesters miR-99b-5p into a duplex form. Since these LNA-based duplexes are resistant to exonuclease activity, they are not degraded within the cell. During RNA isolation for qPCR, these duplex forms are maintained because the RNA is not denatured when preparing the cDNA. Consequently, the miR-99b-5p is less accessible for reverse transcription, resulting in lower detected levels compared to the scramble control-treated samples in the subsequent qPCR reaction.

We hypothesize that this sequestration effect could be responsible for the observed decrease in miR-99b-5p levels measured by qPCR.

In line with this view, numerous studies reported that the administration of a LNA-based anti-miR resulted in decreased detection of the targeted miR via qPCR (some examples are: PMID: 32788885, 34458001, 26936095). Moreover, there are studies specifically addressing the issue related to anti-miR administration and the detection of the corresponding target microRNA via qPCR or Northern Blot (PMID: 23358900).

To further test the aforementioned hypothesis - at least in part - for miR-99b-5p, we performed an experiment using HEK293 cells that exhibit very low endogenous expression of miR-99b-5p. These cells were treated with *miR-99b-5p* mimic to cause overexpression of *miR-99b-5p*. At the same time, these cells were treated with the *miR-99b-5p* inhibitor. Our aim was to test if a qPCR analysis would allow us to detect the reduction of the miRNA mimic by treatment with the miRNA inhibitor LNA. Our results show that the levels of detectable miR-99b-5p increase upon treatment with *miR-99b-5p* mimic, while reduced detection is observed when the cells are co-transfected with miRNA-99b-5p mimic and inhibitor (See novel Fig EV3 and the image below, panel A).

We acknowledge that these data do not fully resolve the question if the anti-miR-99b leads to a degradation of miR-99b-5p or renders it inaccessible to cDNA synthesis and subsequent qPCR detection. However, these additional data show that in our experimental settings, administration of anti-miR-99b leads to reduced detection of miR-99b-5p via qPCR. Since we also demonstrate the effect of anti-miR-99b on specific targets and cellular functions (see Figs. 3 - 5 of the revised manuscript), we are confident that administration of the anti-miR-99b inhibits miR-99b-5p function.

To further address this issue, we also performed the experiment that was suggested by this reviewer. We treated primary microglia cells with the miR-99b inhibitor or control and conducted qPCR analyses at 48 hours (the time point we had used to perform RNAseq; see Fig 3A) and 5 days post-treatment. The results corroborate the initial observations in that significantly lower miR-99b levels were detected via qPCR in cells treated with miR-99b inhibitor when compared to control (See novel Fig EV3 and the image below, panel B).

Figure EV 3 A, B: Administration of anti-miR-99b-5p reduces the levels of miR-99b-5p detectable via qPCR. **A.** To test if the administration of anti-miR-99b oligonucleotides reduces the levels of miR-99b-5p detected via qPCR we treated HEK293 cells - which show low endogenous expression of miR-99b-5p - with either control oligonucleotides (sc-control), miR-99b mimic, anti-miR-99b or the combination of miR-99b mimic and anti-miR99b. While administration of the miR-99b mimic significantly increased the detection of miR-99b-5p via qPCR when compared to sc-control ($P = 0.001$, t-Test), the co-administration significantly reduced detectable miR-

99b-5p ($P < 0.0001$, T-test). $N = 3/\text{group}$. Measurements were performed 48 after treatment. **B.** Primary mouse microglia (DIV 8) were treated with sc-control or anti-miR-99b for either 48h or 5 days before RNA was collected for qPCR. At both time points anti-miR-99b treatment significantly reduced the detectability of miR-99b-5p ($n = 4/\text{group}$, t-test). Error bars indicate SEM.

In summary, it is important to note that LNA-based anti-miRs are known to sequester the target RNA rather than causing the degradation of the target miR. Therefore, the reduced detection of miR-99b-5p upon anti-miR-99b treatment is likely due to the formation of exonuclease-resistant duplexes that are not denatured during cDNA synthesis. Consequently, the miR-99b-5p is less accessible for reverse transcription, resulting in lower detected levels compared to the scramble control-treated samples in the subsequent qPCR reaction. Nevertheless, qPCR is a suitable method to assay target engagement of anti-miR-99b-5p

We have now summarized all of this data as a novel Figure EV3 A, B and its legend and call out this figure on page 6, lines 1-16. Changes are underlined.

Reviewer 1 comments on Figure 2 and says:

“Figure 2: Distance traveled, Time spent in center/ open arms, and pre-pulse inhibition of the startle response are all signs of anxiety. So there is no evidence from experiments in animals that connects miR-99 and SZ. Accordingly, the statement ‘decreased expression of miR-99b-5p is causatively linked to the development of SZ-like phenotypes.’ Is incorrect (line 170) ‘Prepulse inhibition in patients with non-psychotic major depressive disorder, Perry et al, J. affect Disord, 2004”

Although our statement was formulated as a hypothesis we acknowledge that this sentence could be misleading. We have now rephrased it:

“Before performing mechanistic studies, we decided to employ mice in a model system to test the hypothesis that inhibition of miR-99b-5p could lead to behavioral alterations in mice, including SZ-like phenotypes such as paired-pulse inhibition of the startle response (PPI).”

Please see page 6, lines 12-14 of the revised manuscript.

This reviewer also states that we only measured “*signs of anxiety*”.

We understand the reviewer's viewpoint, but our assessment leads us to a different conclusion. As described in the previous version of the manuscript, the behavior tests we apply have been associated with schizophrenia, and especially PPI is an established and widely accepted behavior test to detect schizophrenia-like behavior in mice.

However, it is of course true that behavioral changes in mice cannot fully recapitulate the complex behavioral phenotypes associated with schizophrenia in humans. To avoid any misunderstanding we address this issue more specifically and have carefully revised the entire corresponding paragraph in our manuscript. For example, we have now added the following statement:

“Although these data are encouraging, it has to be mentioned that mice cannot fully recapitulate the complex phenotypes associated with schizophrenia in humans. Thus, our results provide further evidence for a role of miR-99b-5p in schizophrenia but cannot fully establish a causal relationship.”

Please see page 6, last two lines and page 7, lines 1-2 of the revised manuscript. Changes are underlined.

Reviewer 1 comments on Figure 4 and says:

“Figure 4: the authors provide insufficient data to make the case that miR-99b-5p can directly regulate Zbp1 levels (line 278). Prediction algorithms including Targetscan, the most established target prediction algorithm, do not predict Zbp1 downstream of miR-99/100. Neither there is evidence in miRDB. . In fact based on the authors Fig 4b, where a G:U wobble is observed where the continuous seed should be it is more likely that Zbp1 is not a target of miR-99 and it would be a mistake to claim it is. https://www.targetscan.org/cgi-bin/targetscan/vert_80/targetscan.cgi?species=Human&gid=&mir_sc=miR-99-5p%2F100-5p&mir_c=&mir_nc=&mir_vnc=&mirg=

https://www.targetscan.org/cgi-bin/vert_80/view_genetable.cgi?rs=ENST00000371173.3&taxid=9606&members=&subset=1&shownc=1&shownc=1&sortText=cs
<https://mirdb.org/cgi-bin/search.cgi>

Authors should justify their choice and substantiate the evidence for true target:miRNA regulation. The authors should demonstrate an accepted, continuous, miR-99 binding site at the 3'UTR of Zbp1. Then mutate the seed binding site and demonstrate abrogation of regulation. In brain tissue they should provide evidence for direct engagement of Zbp1 with miRNA regulation at the same sequence. For example by data mining of published Ago2 CLIP studies. Other options include unambiguous identification of mirna:target site interactions by different types of ligation Reactions <https://www.ncbi.nlm.nih.gov/pmc/articles/PMC4181535/>

In summary, reasonably well-supported conclusions: miR-99 affect Zbp1 levels and anxiety-like behavior in mice and in microglia

Not sufficiently supported are the hypotheses that miR-99 is specifically involved in SZ and that miR-99 is targeting Zbp1 directly. "

We appreciate these insightful comment and address them one by one below:

1. Target Prediction: Although *Zbp1* is not identified as a target in databases like TargetScan and miRDB, our target prediction was performed using miRWalk, which identified *Zbp1* as a putative target of miR-99b-5p. We completely agree with this reviewer that the prediction of a target can only be a first step that requires further experimental work. Especially, since non-canonical binding sites also exist which might be missed by predictions. This is why we performed luciferase assays. To avoid any misunderstanding we specifically address this issue in the main text. Please see page 9, line 12 - 13, and lines 25 - 28 of the revised manuscript. Changes are underlined.

2. Mutation Studies: The luciferase assays presented in former Figure 4B demonstrate that administration of miR-99b-5p reduces luciferase activity when the luciferase is attached to the 3' UTR of *Zbp1*. This is a well-established experimental approach to test microRNA target engagement. We agree with the reviewer that additional experiments could further substantiate these findings. Therefore, we mutated the miR-99b-5p seed region of *Zbp1*'s 3'UTR. Our findings confirm that addition of miR-99b-5p reduces luciferase activity when using the wild type *Zbp1* 3' UTR in a luciferase assay, but not when using the mutated 3'UTR. We have added this novel data to Fig. 4B and adapted the figure legend accordingly. These data are also described on page 9, line 32 and the methods section on page 24, lines 27-28 of the revised manuscript (Section: *Luciferase assay*). Changes are underlined.

3. Biochemical Context: Although we haven't utilized Ago2 CLIP, we are confident that the data we now provide in the revised manuscript provide strong evidence for the interaction between miR-99b-5p and *Zbp1*'s 3'UTR.

The reviewer also repeats his/her concern that miR-99b-5p may not play a role in schizophrenia but rather in anxiety diseases. We understand that this statement is related to the comment made before. Therefore, please see our response to **Reviewer 1's comment on Figure 2**.

In addition, we like to reiterate that we initially found miR-99b-5p as a candidate microRNA that is deregulated in schizophrenia patients. Our findings demonstrate that miR-99b-5p levels are altered in both blood (using to the best of our knowledge the largest smallRNA dataset so far) and brain tissue of schizophrenia patients. However, we acknowledge that correlation does not necessarily mean causality and thus, we used model systems to further study the role of miR-99b-5p.

Having this said, at present, we cannot exclude that miR-99b-5p may play a role in other neuropsychiatric diseases and we address this issue now in the revised manuscript. Please see page 14, lines 28-30. Changes are underlined.

Reviewer 1 has several minor comments.

Minor comments

Writing of the main text with terms that refer to graphics of the figure (lines 113-114 "The turquoise, yellow, blue and red" should be changed to meet standard practice. Line 128 "Taken together, these data suggest that especially the 128 microRNAs present in turquoise, pink, green, blue, yellow and red modules warrant further analysis." This is completely obscure to the reader. Re-write with scientific-relevant conclusions

Referring to cell free miRNA as "expression" is wrong; as these might be spilled out of cells and in any way do not obey "regulation of gene expression" terms. Simply use "levels".

In response to these comments, we have now revised the main text accordingly. Please see page 4, lines 9 – 34, and page 5, lines 1 -7 of the revised manuscript. We also made sure that we do not refer to microRNA expression but “levels” when discussing the data obtained from blood.

Line 159-161 Expect for miR-500a-159 3p and miR-221-5p, the expression of all other candidate miRs have is significantly correlated to the 160 PANSSs, GAF and BDI-II scores" delete "have"

We have deleted the word “have” in this sentence.

How "GO analysis of the downregulated genes did not yield any highly significant pathways but detected processes linked to voltage-gated potassium channels (line 194) is suggesting immune-related processes (line 196) should be better explained.

We agree that this sentence could be misleading. We rewrote this sentence accordingly. Please see page 7, lines 7 - 11 of the revised manuscript. Changes are underlined.

Refernce 34 as link of microglia to SZ is weaj (R. Rahimian, M. Wakid, L. A. O'Leary, and N. Mechawar, "The emerging tale of 996 microglia in psychiatric disorders.," *Neurosci Biobehav Rev.*, vol. 131, pp. 1-29, 2021.) and should be substantiated by better more central citations of literature.

We agree with this. During the reviewing and revision process, a comprehensive and more timely review article has been published on this topic by Zhou *et al.* 2023 in the Journal *Schizophrenia*. We are aware that the citation of a review article may raise concerns since not enough credit is given to the primary literature. However, it is impossible to cite all of the key findings in this area. Thus, in addition to this review, we cite now one of the earliest studies showing the activation of microglia in schizophrenia patients. In fact we had already cited this study by Bayer *et al.*, 1999, but not in the context of this sentence. In addition, we cite a study by Breitmeyer *et al.*, which to the best of our knowledge reflects the most recent data relevant to this topic in the context of our work. Please see page 8, line 3. Changes are underlined.

After 3h the authors state that microglia lacking miR-99b-5p may have detrimental effects on synaptic function when co-cultured with cortical neurons (line 245). However, the needed evidence for that is only in Fig 3i so the conclusions should be positioned after 3i.

We refer to this finding also after Fig.3i. The sentence in question was aimed to formulate the corresponding hypothesis. We understand that this might have been misleading and have deleted this sentence after the reference to Fig 3i.

Rephrase broken sentence: GO term analysis was performed for the 13 genes and revealed that they are linked to inflammatory processes including type I interferon signaling (Fig. EV7, supplemental table 14), linked to schizophrenia [37] [38]. (lines 263,264).

We rephrase this sentence. Please see page 9, line 15 of the revised manuscript. Changes are underlined.

Discussion is long (lines 316-457) and can benefit from editing and focusing

This was also pointed out by reviewer 3 as one of the minor comments. We carefully reevaluated the discussion to be more concise. Please see also the corresponding comment by reviewer 3 who suggested to specifically shorten our discussion on the other candidate microRNAs in addition to miR-99b-5p.

Referee #2:

Comments:

In this study, Kaurani et al. identify a miRNA -miR99b- that is down-regulated in Schizophrenia (SZ) patients and may play an important role in the pathogenesis of SZ by promoting pro-inflammatory responses via its target Zbp1. The work stems from the sequencing of the MicroRNAome from blood and brain samples of two different sets of SZ patients and then focuses on dissecting the molecular pathway downstream of miR99b and its relevance for SZ.

To this goal, the authors use a combination of in-vivo and in vitro approaches, spanning from mice behavioral experiments to bioinformatics analyses and tests of microglia activity. Overall, the manuscript is interesting and can count on two major strength points. Firstly, the data gathered from the patients provide a precious resource for further studies, and their analysis a useful guideline on how one could integrate different types of biological information. Secondly, the body of evidence collected by Kaurani and colleagues in favour of a role for miR99-b and Zbp1 in SZ, summed to the fact that this pathway has never been studied before in the context of SZ, makes the study of relevance to deepen our understanding of a disease still orphan of effective treatments such as SZ.

Nevertheless, the enthusiasm towards the work as a whole is dampened by the presence of some experimental issues which need to be addressed to meet the standards of EMBO J. In addition, we think that the presentation of the data, quality of figures and writing could also be improved. Below we detail our comments, hoping that this helps the authors to improve the manuscript.

Main points:**Reviewer 2 says:**

“The main claim of the paper - the microglia-mediated effects of miR-99b downregulation in SZ - is mostly grounded on in-vitro experiments which are overall well-designed and executed. Nevertheless, we think that some of the in-vitro results need to be substantiated by further in-vivo evidences. To us it is unclear, for example, why the authors did not investigate the presence of pro-inflammatory changes and/or morphological changes like spine density in the brains of mice injected with the anti miR-99b.”

We appreciate this comment. We have addressed - in part - the molecular changes upon miR-99b-5p knock-down in mice (in vivo) within the data presented in former Fig. 2g - j. It is true that we did not investigate spine density and synapse number in the brains of mice injected with anti-miR-99b. At present, we do not have tissue sections from anti-miR-99b-injected mice at hand, which would be important to perform for example immunohistochemical analysis. Thus, we would need to repeat the entire experiment, which will require us to submit a novel animal experimental protocol and wait for approval. Especially, when taking into account the way the 3Rs are currently interpreted by the state authorities in Germany, at present this is not possible. However, taking into account that we provide data from patients and animal models that point to the involvement of miR-99b-5p in schizophrenia and more specifically in microglia, we are confident that our functional analysis in primary mouse cells as well as human iPSC-derived microglia provides sufficient evidence for the role of miR-99b-5p in regulating microglia function and neuronal plasticity, which we have analyzed via RNA-seq, immunohistochemistry and electrophysiology.

We also like to refer to a comment made by reviewer 3, who asked us to study dendritic spine density not only upon miR-99b-5p inhibition but also in response to co-administration with *ZBP-1* ASOs. We were able to perform this experiment and the data is now shown as novel panel L within Fig. 3. Please see our response to reviewer 3.

Reviewer 2 continues:

Furthermore, to what extent miR-99b expression is cell-type specific is not evident from the data provided, and could be readily addressed by combining imaging technologies for detection of microRNAs with cell-type markers labeling. The same is true also for the downstream target *Zbp1*, for which no details are provided concerning its expression in the PFC, with the exception of the qPCR of figure EV8a.

We agree that it would be interesting to learn more about the general expression pattern of *miR-99b-5p* and *Zbp1*. To this end we have now employed MACS to isolate astrocytes, oligodendrocytes and microglia from the prefrontal cortex of adult mice and analyzed miR-99b-5p as well as *Zbp1* expression via qPCR.

In summary, within the mouse PFC miR-99b-5p is expressed in microglia, astrocytes and oligodendrocytes, while *Zbp1* is specifically expressed in microglia. We provide this data within the novel Appendix Fig S4. The data is described on page 9, lines 34 and page 10, lines 1 - 3 of the revised manuscript. Changes are underlined.

Reviewer 2 says:

- At line 117-118, the authors state that the clinical phenotypes PANSS and BDI-II are decreased in SZ patients and GAF is increased. However, from the graphs in figure EV1 it appears clearly the opposite: PANSS and BDI-II are increased in SZ patients and GAF is decreased. Moreover, on the graph plotting the PANSS score the group labels (SZ and control) are inverted.

Thank you for pointing out these discrepancies. The PANSS and BD-II scores are of course increased in patients, while the GAF score is decreased. We have now corrected this in the revised manuscript. Please see page 4, lines 24 - 25. Changes are highlighted.

We also corrected the labeling of Figure EV1.

Reviewer 2 says:

From the DE analysis of SZ patients the authors identify 93 (blood) and 68 (brain) differentially regulated miRNAs (fig 1e, 1f). These numbers are at odds with what reported in fig 1g, where a total of 34 miRNAs from blood and 9 from brain is reported. Can the authors clarify?

We agree that our description was confusing. To clarify this issue we like to state the following.

- (1) It is true that we identified 93 differentially expressed microRNAs in blood. Of these, 34 miRs were down-regulated. These data were reported in Supplementary table 4 (now Dataset EV4). The Venn diagram in Fig 1G only shows data from down-regulated microRNAs, hence the 34 microRNAs downregulate in blood.
- (2) In the previous version of the manuscript, we reported in the text 36 microRNAs that were decreased and 32 that were up-regulated in the postmortem brains of SZ patients. We noticed now that this was a mistake. The correct numbers were reported in supplementary table 5 (now Dataset EV5) but have been confused in the main text. The correct numbers are 28 down-regulated and 31 up-regulated microRNAs. We have corrected this in the text of the revised manuscript.
- (3) In the Venn diagram of Fig 1F there is indeed a mistake. We believe that this is an editing mistake since the numbers within the Venn diagram were not easily readable in the image generated by the software. This is why we changed the font and size in Adobe Illustrator. We assume there has been a typo. We have now double-checked the numbers and confirmed that the updated Venn diagram matches the data in the supplemental tables.

To avoid any confusion, we also rewrote this part of the analysis, changed the Venn diagram in Fig 1F accordingly and also show a Venn diagram of the up-regulated microRNAs separately as a novel panel within Expanded view Fig. 2

We hope that the revised version of the text and the additional data helps to avoid any misunderstanding. Please see novel panel A of Fig EV2 and the corresponding legend as well as the main text on page 5., lines 18 - 33.

Reviewer 2 states:

“- Related to the point above, where is the demographic information for the human blood samples? Were samples age-, sex- and ethnicity-matched, for example?”

A summary of the demographics was presented in Supplementary table 1 (Now Dataset EV1). In our analysis, we corrected for age and sex and had referred to this in the methods section. Please see page 16, line 20 of the revised manuscript.

Regarding ethnicity, 98,01% of the participants were of European origin. We also report this info in the revised version of the manuscript on page 15, line 11.

Reviewer 2 says:

“- It appears that while for the behavior a LNA-based miR-99b-5p was used, an ASO-based inhibitor was used for the RNA sequencing data. This is problematic on several fronts. First, the data generated this way cannot be directly compared. Second, what is the evidence that the two inhibiting systems perform equally well? At the very least, the authors have provide evidence for equipotency of inhibition of the two approaches, or better, repeat one of the experiments with the other inhibitor. “

This is a misunderstanding. We used LNA-based anti-miR-99b oligonucleotides for all experiments in which we inhibited miR-99b-5p function. ASOs were used to knock down *Zbp1*.

We apologize if this was not clearly communicated in our manuscript and have now revised the corresponding sentence in the methods section to avoid any confusion. Please see page 17, lines 28 - 31 of the revised manuscript. Changes are underlined.

Reviewer 2 says:

“- Related to the above, for the anti-miR99b experiment of figure 2 and the *Zbp1* ASO rescue experiment of figure 4, did the authors verify surgery efficiency -ie. check that only PFC is targeted, that this is comparable amongst the different mice groups? Moreover, did they check the levels of *Zbp1* expression itself following the ASO manipulation for both the in-vitro and the in-vivo experiments? “

To analyze the specificity of anti-miR-99b and *Zbp1*-ASO injections to the prefrontal cortex (PFC), we had also isolated the hippocampal region followed by qPCR analysis for miR-99b-5p and *Zbp1*. This nearby brain region should not be affected by the PFC injection. In contrast to our analysis of the PFC, no significant changes in the detection of miR-99b-5p or *Zbp1* were observed in the hippocampus when compared to the control group. These novel data are now presented as panel C within Fig. EV3 and described on page 6, lines 20- 21 and page 11, line 6 - 7 of the revised manuscript. Changes are underlined.

To address the second part of the questions, we performed a validation experiment using primary microglia cultures. Upon treatment with LPS we observed a highly significant upregulation of *Zbp1*, which was significantly attenuated when microglia were treated with *Zbp1* ASOs. These data are

now presented as a novel Appendix Fig. S5 and are described on page 10, lines 18 - 19 of the revised manuscript. Changes are underlined.

Reviewer 2 remarks

“- In the experiment of figure EV4, to which sample does the FACS plot refers to? Please provide plots for both experimental groups, as well as the negative control used for setting the gates (eg. Isotype control). Moreover, it is unclear how the microglia cell numbers in EV4 b were determined. We suggest an immunohistochemistry approach to achieve a more direct quantification. “

We have now revised Fig EV4 (now Appendix Fig S2) accordingly and show FACS plots for both experimental groups. The cell numbers were calculated as % microglia as percentage of living cells. An isotype control was performed along with the experiment but the corresponding images were not saved. However, these experiments were performed by our collaborators and co-authors from the Flügel lab (JS, FO and JS). The group has ample experience with immune cell sorting and is routinely performing CD45-Cd11b staining which gives always a very clear population of cells (e.g see Hosang L, et al. Flügel A, Odoardi F. *Nature*. 2022 Mar;603(7899):138-144; Berghoff, S.A.,... Odoardi... et al. et al. *Nat Neurosci* 2021 24, 47–60 or Schäffner E....Flügel..et al. *Nat Neurosci*. 2023 Jul;26(7):1218-1228)

We also appreciate the suggestion to quantify microglia cells via immunohistochemical (IHC) analysis. IHC has the advantage of spatial information but is - in contrast to FACS - semi-quantitative. Thus, we are confident that the quantitative FACS analysis is better suited for the experimental questions addressed here.

- In some cases, the statistical tests chosen are not coherent with the way data are plotted. For example in figure 2a, where two independent variables are present (days and treatments) on the same graph. The data should be analyzed using a two-way ANOVA, and mir-99b expression values be expressed relative only to the 5 days control timepoint. Yet mir99b values at each timepoint are plotted relatively to each matched control (5 days and 10 days), and t-test is used. The same happens in figure 2j. Data are analyzed by t-test, but plotted on the same graph. In other cases, the statistical information is missing / not clearly stated. For example, is the difference in Zbp1 expression detected by WB following anti miR99b significant (figure 4c)? Also, from the legend is not clear if a t-test was used for data in figure 4f-4k. If that's the case, it would have not been more appropriate to perform a one-way anova, since 3 groups were compared? Finally, could the authors clarify how the PPI data from figure EV8 were analyzed? From the legend, it seems that they used a combination of one-way Anova, two way Anova and t-test. This approach seems unusual and it is at odds with the other PPI experiment shown in figure2, where only t-test was used.

We apologize for the confusion and have now clarified the use of statistical tests. In brief:

Fig. 2a: The analysis of miR-99b-5p levels 5 and 10 days after injection are two independent experiments. Our mistake was that we plotted both datasets in one graph. This could be misleading and we have now corrected this.

Figure 2j: We made the same mistake as in Fig 2a in that we plotted all data in one graph, although these are three independent experiments. We have corrected this now in the corresponding figure panel.

Figure 4c: We apologize for this mistake. These data were analyzed using a t-Test. There is a significant increase of ZBP1 protein levels in cells treated with anti-miR-99b when compared to sc-control. We corrected this within the novel version of Fig.4c and the corresponding figure legend.

Figure 4f-k: For these data, we had indeed performed a one-way ANOVA first, followed by a t-Test for post-hoc analysis in case the ANOVA revealed a significant difference amongst groups. We missed to report this. The asterisk in the previous version of the manuscript only hinted at the results from the t-Test. We have now corrected this in the corresponding figure legend.

Fig EV8 now Fig. 5: This seems to be an editing mistake since we did not use a Two-way ANOVA to analyze the data. We have now carefully checked the legend for Fig 5b and corrected it.

Minor points:

- We find that the manuscript is at places of a difficult read. Many paragraphs are written as a concatenation of information, and not well structured. This should be improved as it will ultimately help the reader (and reviewer). Here are the most worrisome examples: A) spelling/grammar mistakes (eg. line 159, line 329) B) the use of expressions not precise enough for a scientific article (eg. line 41 "vast majority"; line 65 "most of the"; line 172 "representing"; what is the difference between "primary microglia" (line 217) and "primary microglia" (line 230)? The way it's written, it seems that these are two different experiments, while in reality they are not: line 364 "essentially nothing was known") C) Sentences juxtaposed using elusive logical implications (eg. the word "however" on line 41 is not the right connecting word, as the information presented in this sentence is not contradicting the information provided in the previous one. (On line 65, in contrast, "however" is used the right way, as both sentences are referring to the same type of information.); from line 80 to 89, from line 320 to 322, from 424 to 426.) D) A substantial part of the discussion section is dedicated to exhaustively reviewing the literature on the role of other miRNAs in SZ, when other highly relevant topics (eg. the mechanisms causing miR-99b and/or miRNAs deregulation in general in SZ, state of the art of miR-based therapeutic/diagnostic strategies in psychiatric disorders and/or possible approaches for rescuing miR-99b expression and/or block Zpb1 in SZ) are dealt with in few lines or not touched at all.

We really appreciate these helpful comments. We have now addressed all of the points specifically raised in this comment. We also modified the discussion accordingly with the aim to make it appear less like a literature review.

- For the sequencing experiment of figure 2g, were the animals pooled together? If yes, how many mice per group? Moreover, how many replicates of the RNA-seq data were used to do the DEG analysis? Please provide these details in the main text and/or figure legend.

While we make the raw data available via the GEO database, we apologize that we missed communicating this information. We used n = 3 per group for RNAseq. The data is from individual animals and was not pooled. We now refer to this in the figure legend.

- We find that the general layout of the figures as well as the level of information provided in the figures and/or figure legends could be improved to be more intuitive and easier to interpret. A few examples A) Figure 1. In 1b-c, for some clusters it is possible to see the individual data points, for others is not. Please make it consistent. Also, the abbreviation ME (= expression module) is not clearly spelt out in the legend, and the technical term "eigen-expression" is not explained either. This makes the first part of figure 1b not very intuitive. Perhaps the authors could include in the main figure 1 or in the supplementary a scheme for the bioinformatic analysis encompassing the results of 1b-1d. Finally, the graphs in 1e and 1f could benefit from a title and a legend.

We have now changed the corresponding figure panels in Fig. 1 and the corresponding legend as suggested.

We explained the abbreviation ME in the methods section "WGCNA analysis". To further clarify, we now expanded this paragraph and also explain here in more detail the concept of eigen-expression. Please see page 17, lines 4 - 6 of the revised manuscript. Changes are underlined.

B) Figure 2. It is difficult to see the individual data points for 2b-2f and 2j. Please change colors for the anti miR-99b group. For 2g, could the authors indicate which dots correspond to *Il1b*, *Tgfb1* and *Tnfa*, since they claim in the main text that these genes are differentially expressed. As for 1e and 1f, and 2g, they could be improved with title and legend.

We changed the figure panels accordingly. All data points are now easily visible.

Regarding the volcano plot in Fig 2g, we realized our earlier explanation of the data was misleading. The qPCR results for *Il1b*, *Tgfb1*, and *Tnfa* in Fig 2j don't confirm the RNA-seq findings in Fig. 2g. These genes exhibited differential expression in primary microglia and iPSC-derived human microglia (See former supplemental tables 9 and 11 now Datasets EV9 and 11 and Fig 4). Also, upon miR-99b-5p knockdown in the mouse PFC and subsequent microglia isolation via FACS, these genes showed significant changes, as depicted in Fig. 2j. In Fig 2g, where bulk tissue datasets are presented, these three genes showed an increase but didn't reach statistical significance. Our RNA-seq analysis in Fig 2g suggested a specific effect on microglia (based on Fig 2h & i). To delve deeper, we conducted qPCR on genes associated with microglia-mediated inflammatory processes. We hypothesized that genes with "borderline significance" in bulk data might show significance when analyzed specifically in microglia. This is why we tested - amongst other genes - also *Il1b*, *Tgfb1*, and *Tnfa*. Genes that were differentially expressed in the dataset shown in Fig 2g could also be confirmed in FACS-isolated microglia. Here, we performed qPCR for *Zbp1* that differed significantly in control and anti-miR-99b samples in both bulk tissue when analyzed via RNAseq and in FACS-isolated microglia via qPCR. We didn't include *Zbp1* in Fig 1j due to detailed elaboration later in the manuscript. We acknowledge this oversight might be misleading. We've now revised the data description and included *Zbp1* qPCR data in Fig 2j and explain this experiment and the data in greater detail. Please refer to the updated Fig 2 and page 7, lines 24- 32 of the main text for the underlined changes.

C) Figure EV6. Plotting all the qPCR data for different genes on the same graph is misleading. The authors should at least indicate for each pairwise comparison the gene measured, as done in figure EV6a.

We have changed these figure panels (now Fig EV5) accordingly.

D) Figure 3i: Absolute values of dendritic spines must be given per given segment. It is not clear what the number of dendritic spines was normalized to, nor the statistical test used.

In the literature, researchers often present spine density normalized to control in case of perturbation experiments as we did in our study. It's also true that spine density is presented as "absolute values", for example, most often as spines/10 μ m dendrite. This is in fact how we originally analyzed our data. Nevertheless, we decided to present the data as normalized values, so that the effect size can be more easily compared to other studies because the absolute number of spines per dendrite can differ due to culture conditions. However, we are also happy to provide the data as spines/10 μ m dendrite and have now updated panel L in Fig 3 accordingly.

We also now describe in greater detail the statistical test used for analysis. Please see the revised legend of Fig. 3. Changes are underlined.

- Lastly, the discussion is too long and off target. For example, all the text pertaining to microRNAs other than miR-99b-5p can be cut.

We have now carefully revised the discussion as suggested by this reviewer. We specifically aimed to make the discussion more concise.

Reviewer #3:

The study by Kaurani et al. identifies the miR-99b-5p-Zbp1 pathway as contributing to the pathogenesis of schizophrenia. Specifically, their results show that levels of miR-99b-5p are reduced in blood and postmortem brains from schizophrenia patients while pro-inflammatory genes (including Zbp1) are upregulated. To establish a causal link between miR-99b-5p and schizophrenia-related behaviors, they went on by inhibiting miR-99b-5p in the prefrontal cortex of 3-months old mice and found that this led to schizophrenia-related phenotypes and was associated with an increase in the expression of pro-inflammatory genes that are enriched in microglia. The authors could confirm those results in in vitro preparations and further demonstrated that inhibiting miR-99b-5p in microglia leads to increased phagocytosis and decreased spine number and was again associated with an upregulation of genes associated with neuro-inflammatory processes and synaptic pruning. From their RNAseq data, the authors identified the pro-inflammatory gene Zbp1 as a direct target of miR-99b-5p. Importantly, they could demonstrate that silencing Zbp1 rescues the effects of inhibiting miR-99b-5p on caspase activity, IL1beta expression and phagocytosis in both primary or human iPSC-derived microglia. Importantly, they finally demonstrate that silencing Zbp1 also rescues behavioral deficits induced by inhibiting miR-99b-5p in the prefrontal cortex of mice.

Overall, this is an important, impressive and elegant study that combines molecular manipulations to high throughput transcriptomic and refined bioinformatics analysis in both in vivo and in vitro models. The experimental approach is elegant and compelling and the conclusions are well supported by the data. The fact that many of the results could be recapitulated in both human and mouse models, both in vivo and in vitro underscores the importance and robustness of the findings. Not only do the authors identify a new signaling pathway that likely contributes to schizophrenia pathogenesis but they also validate other miRNA candidates that were previously linked to schizophrenia and other brain disorders. I would recommend the paper for publication provided that the following minor concerns are addressed:

Reviewer #3 says

“•P8-L274 - 'Co-transfection of this vector with miR-99b-5p LNAs significantly reduced the luciferase activity (Fig 4b), but this was not the case when scramble control LNAs were used'. I believe that in the luciferase assay, the luciferase reporter construct carrying the Zbp1 UTR or a scramble (control) UTR was co-expressed with a miR-99b-5p mimic, not the miR-99b-5p LNA. This should be corrected.”

This is true. We corrected this mistake. The sentence now reads : “Co-transfection of this vector with miR-99b-5p mimic significantly reduced the luciferase activity (**Fig 4b**), but this was not the case when scramble control were used or when the miR-99b-5p seed region was mutated (Fig 4b).” Please see page 9, lines 30 - 32 of the revised manuscript.

Reviewer #3 says

“•Fig. 3E - The authors should illustrate the phagocytosis assay with representative images.”

Similar to former Fig EV6 (now Fig. EV4) we have now added representative images to Fig. 3e.

Reviewer #3 says

“•Fig. 3i - It would be nice to test whether Zbp1-ASO could also rescue the decrease in spine density to also have a cellular readout of the regulation of Zbp1 by miR-99b-5p at the neuronal level. “

We have now performed this experiment as suggested. We observed that the administration of Zbp1-ASOs was able to rescue the anti-miR-99b-mediated decrease in spine density. These data are depicted as novel panel L within Fig. 4 and are described on page 10, lines 31 - 33, and page 11 lines 1 – 2 of the revised manuscript.

Reviewer #3 says

“•The strategies used by the authors to inhibit miR-99b or Zbp1 is not specifically targeted to microglial cells. The authors should thus discuss in more details the relative expression of miR-99-5p in microglia versus neurons or other glial cell types and discuss their results in light of a potential action of anti-miR99b or Zbp1-ASO in cells other than microglia. “

This question is similar to the second comment by reviewer 2. We agree that it would be interesting to learn more about the general expression pattern of miR-99b-5p and Zbp1. To this end, we have

now employed MACS to isolate astrocytes, oligodendrocytes and microglia from the prefrontal cortex of mice and analyzed miR-99b-5p as well as *Zbp1* expression via qPCR.

In summary, within the mouse PFC miR-99b-5p is enriched in microglia when compared astrocytes and oligodendrocytes, *Zbp1* is specifically expressed in microglia. We provide this data within the novel Appendix Fig. S4. The data is described on page 9 lines 34 and page 10, lines 1 - 3 of the revised manuscript. Changes are underlined.

In response to this question, we also discuss in greater detail that we cannot exclude the possibility that other cell types contribute to the observed phenotypes in vivo. Please see page 14, lines 30 - 33. Changes are underlined.

Reviewer #3 says

“•I find the data shown in Fig. EV8 b,c important as it demonstrates the role of the regulation of Zbp1 by miR-99b-5p in schizophrenia-related behaviors. I would recommend to show them in a main figure.”

As requested, former Fig. EV8 panel b and c is now shown as main Fig. 5.

Dear Prof. Fischer,

Thank you for the submission of your revised manuscript to The EMBO Journal. We have now received the comments of the three referees that were asked to re-assess your study (included below). As you will see, the referees mention that the study is novel, interesting, and convincing, and they all recognize that the revised manuscript is significantly improved, and that their previous concerns have been satisfactorily addressed. There is only one remaining point of referee #2 referring to their previous comment 7 (i.e. need for showing expression levels of both the miRNA and Zbp1 in the targeted and control regions) that we would like you to address in a revised version of your manuscript before we can proceed with its acceptance for publication. Please include in your re-submission a brief response to this comment explaining how it is addressed in your revised manuscript, and also correct the typos and mistakes that referee #2 identified.

From the editorial side, there are some changes and corrections that we need from you before we can proceed with handling of the manuscript:

- We noticed that your manuscript has an unusually high number (5) of co-corresponding authors. Please consider whether it can be reduced or explain in the cover letter of your re-submission if this is necessary and justified based on their actual contributions.
- Please note that all co-corresponding authors are required to supply an ORCID ID. Please see our Authorship guidelines for more information and instructions on how to link your ORCID ID to your account in our manuscript tracking system: <https://www.embopress.org/page/journal/14602075/authorguide#authorshipguidelines>.
- Please enter all relevant funding information in our online manuscript handling system (eJP). It should match exactly the information provided in the Acknowledgements section of your manuscript. We have noticed the following inconsistencies in your current submission: missing in eJP: GRK2824; ERA-NET Neuron Project EPINEURODEVO and the JPND project EPI-3E; DFG, German Research Foundation, project number 514201724; Studienstiftung des Deutschen Volkes and the International Max-Planck Research School for The Mechanisms of Mental Function and Dysfunction (IMPRS-MMFD); Chan Zuckerberg Initiative Neurodegenerative's Challenge Network (2020-221779(5022) & 2021-235147); International Max-Planck Research School for Genome Science; Max-Planck Research School for Neuroscience; the German Ministry of Education and Research (BMBF; 01EE1404H), and the Dr. Lisa Oehler Foundation (Kassel, Germany); DFG (DFG; FA 241/16-1).
- Please provide a list of up to 5 keywords after the Abstract in your revised manuscript.
- Please change the heading of your Data accessibility section to "Data availability". All deposited data should be publicly available at the time of publication, and the reviewer tokens can therefore be removed from this section now.
- Please update the heading of your Conflict of interest statement to "Disclosure and competing interests statement".
- The author contributions statement should be removed from the manuscript file. Instead, we use CRediT to specify the contributions of each author in the journal submission system. Please use the free text box to provide more detailed descriptions. See also our guide for more information: <https://www.embopress.org/page/journal/14602075/authorguide#authorshipguidelines>.
- We noticed that callouts for Fig. 5A-B are missing. Please make sure that all Figure panels are callout out (in alphabetical order) in your revised manuscript.
- Please state in your Materials and Methods the details of the authorities granting ethics approval -including the respective reference numbers- of the experiments involving human participants and animals, instead of only citing an older reference.
- Please correct the name of the journal in the top section of your Author Checklist ("The EMBO Journal" instead of "EMBO").
- Each Dataset needs to be uploaded as an individual file with its legend in a separate tab in the same Excel file.
- Please include the title of the manuscript on the first page of your Appendix file.
- The Source Data for Fig. 3J appears to be missing. Please provide the missing data or explain/clarify.
- Please note that EMBO press papers are accompanied online by:
 - A) a short (2 sentences) summary of the findings and their significance,
 - B) 2-5 short bullet points highlighting the key results, and
 - C) a synopsis image that is exactly 550 pixels wide and 300-600 pixels high (the height is variable). You can either show a model or key data in the synopsis image. Please note that the text needs to be readable at the final size.

Please upload this information along with your revised manuscript (the text for A and B should be provided in a separate Word file).

- Please note that a separate "Data Information" section is required in the legends of Figures 1b-c; 2a-j; 3d-e, i; 4b-l; EV 4a-b, d; EV 5a-c. For more information and an example, please see our guide: <https://www.embopress.org/page/journal/14602075/authorguide#figureformat>.
- Please note that the Figure panel 4g is not labelled in the Figure, however the corresponding legend for the same is labelled as 4g. This needs to be rectified.
- Please indicate the statistical test used for data analysis in the legends of Figures 2f-i; 3c-e, h; EV 2b.
- Please note that in Figures 2a, e, j; 3i; 4b-l there is a mismatch between the annotated p values in the figure legend and the annotated p values in the Figure file that should be corrected.
- Please note that information related to "n" is missing in the legends of Figures 1e; 2f; EV 4a-b, d.
- Although "n" is provided, please describe the nature of entity for "n" in the legends of figures 1b-c; EV 4a-c.
- Please note that the error bar is not defined in the legend of Figure EV 3c.
- Please note that the measure of center for the error bars needs to be defined in the legends of Figures 2a-f, j; 3d-e, i; 4b-l.
- Please note that scale bar and its definition are missing for Figure EV 4c.
- The main and EV Figure legends should be moved after the References.

As soon as these issues are resolved, I will contact you again to discuss with you a few suggestions for minor textual improvements in the title and abstract.

Please also note that as part of the EMBO publications' Transparent Editorial Process, The EMBO Journal publishes online a Peer Review File along with each accepted manuscript. This File will be published in conjunction with your paper and will include the referee reports, your point-by-point response and all pertinent correspondence relating to the manuscript. You can opt out of this by letting the editorial office know (contact@embojournal.org). If you do opt out, the Peer Review File link will point to the following statement: "No Peer Review File is available with this article, as the authors have chosen not to make the review process public in this case."

We look forward to seeing a final version of your manuscript as soon as possible. Please use this link to submit your revision: <https://emboj.msubmit.net/cgi-bin/main.plex>

Yours sincerely,

Referee #1:

Kaurani et al suggest in "A novel miR-99b-5p-Zbp1 pathway in microglia contributes to the pathogenesis of schizophrenia" that miR-99b-5p is downregulated in both the prefrontal cortex and blood of patients suffering from schizophrenia. Accordingly, inhibition of miR-99b-5p leads to a schizophrenia-like phenotype in mice. miR-99b-5p mediate an inflammatory response in microglia, upstream of Zbp1. Accordingly, Zbp1 knockdown ameliorates the pathological phenotypes caused by miR-99b-5p inhibition.

The discovery of a novel miR-99b-5p-Zbp1 axis in microglia might be relevant to the pathogenesis of schizophrenia.

The work is novel and interesting.

The revised version of the manuscript is significantly improved.

I think it can be accepted for publication.

Referee #2:

In the rebuttal letter, the authors addressed all points previously raised in an appropriate and comprehensive manner, except for comment 1 (further in vivo evidence for miR-99 effects) and comment 7 (surgery efficiency), which were only partially dealt with. In the case of comment 1, we understand the legal and practical difficulties of having to perform a new batch of surgeries, and we appreciate the new in vitro data provided instead. For comment 7, we find the new data from hippocampi of mice that underwent surgery convincing enough for the anti-miR99 experiment. However, for the miR99 + Zbp1 ASO experiment, the authors only show Zbp1 levels in hippocampi. For this experiment, it would be important to show that the expression levels of both miR99 and Zbp1 is down in the region targeted (PFC) and unchanged in the control region (hippocampus). Furthermore, we noticed some typos in the legend for Appendix Fig S5 and in the main text (for example, authors refer to Figure 1F but it should be Figure 1G instead). Finally, we acknowledge that the quality of the figures and readability of the main text has substantially improved. Overall, we considered ourselves satisfied with the responses provided to our previous comments and with the work carried out during this revision.

Referee #3:

The authors did a great job in replying to the referee's concerns raised on the initial version of the manuscript. I have no further comment on this interesting and convincing study.

Response to editorial comments.

1. We noticed that your manuscript has an unusually high number (5) of co-corresponding authors. Please consider whether it can be reduced or explain in the cover letter of your re-submission if this is necessary and justified based on their actual contributions.

We have carefully considered the comment regarding the number of co-corresponding authors and wish to address this question.. The designation of five co-corresponding authors—LK, TGS, PF, FS, and AF—reflects the multidisciplinary and collaborative nature of this study. Each of these authors made substantial contributions to the study, representing specific expertise that justifies their inclusion as co-corresponding authors. There is also a track record of this team that for example jointly signed another highly interdisciplinary study published in *EMBO Mol Med.* as co-authors (Islam et al., 2021; PMID: 34633146).

The PsyCourse dataset serves as the foundation for our study, comprising over 600 human blood samples from deeply phenotyped individuals. Obtained through a large collaborative project, TGS, PF, FS, and AF contributed specific clinical, genetic, phenotyping, and molecular expertise, respectively. To elaborate further, PF implemented the PsyCourse cohort, while TGS analyzed and managed the extensive phenotypic data. FS handled smallRNAome analysis, and together with AF, conceptualized and designed the molecular and functional aspects of the study. Additionally, LK, a senior scientist, played a pivotal role in orchestrating the study, conducting a wide range of experiments, and performing data analysis.

Given the collaborative and interdisciplinary nature of this research, the involvement of multiple co-corresponding authors accurately reflects the collective effort and expertise required for the successful execution of this study, with each author representing a specific knowledge in this joint research endeavor.

2. Please note that all co-corresponding authors are required to supply an ORCID ID. Please see our Authorship guidelines for more information and instructions on how to link your ORCID ID to your account in our manuscript tracking system:

Please find the ORCID of the corresponding authors.

Lalit Kaurani: 0000-0002-2304-6111

Thomas G. Schulze: 0000-0001-6624-2975

Peter Falkai: 0000-0003-2873-8667

Farahnaz sananbenesi: 0000-0002-2123-7694

Andre Fischer: 0000-0001-8546-1161

In the online submission system the ORCID for LK, PF and AF are already included. Could you please help us to add the ORCID for TGS and FS that are provided above.

3. Please enter all relevant funding information in our online manuscript handling system (eJP). It should match exactly the information provided in the Acknowledgements section of your manuscript. We have noticed the following inconsistencies in your current submission: missing in eJP: GRK2824; ERA-NET Neuron Project EPINEURODEVO and the JPND project EPI-3E; DFG, German Research Foundation, project number 514201724; Studienstiftung des Deutschen Volkes and the International Max-Planck Research School for The Mechanisms of Mental Function and Dysfunction (IMPRS-MMFD); Chan Zuckerberg Initiative Neurodegenerative's Challenge Network (2020-221779(5022) & 2021-235147); International Max-Planck Research School for Genome Science; Max-Planck Research School for Neuroscience; the German Ministry of Education and Research (BMBWF; 01EE1404H), and the Dr. Lisa Oehler Foundation (Kassel, Germany); DFG (DFG; FA 241/16-1).

We have now added the missing information to the online manuscript handling system.

4. Please provide a list of up to 5 keywords after the Abstract in your revised manuscript.

We have now added 5 keywords to the revised manuscript after the abstract.

5. Please change the heading of your Data accessibility section to "Data availability". All deposited data should be publicly available at the time of publication, and the reviewer tokens can therefore be removed from this section now.

We have changed the heading accordingly.

6. Please update the heading of your Conflict of interest statement to "Disclosure and competing interests statement".

We have changed the heading accordingly

7. The author contributions statement should be removed from the manuscript file. Instead, we use CRediT to specify the contributions of each author in the journal submission system. Please use the free text box to provide more detailed descriptions. See also our guide for more information: <https://www.embopress.org/page/journal/14602075/authorguide#authorshipguidelines>.

We have removed this section from the manuscript and used the free text box option in the journal submission system.

8. We noticed that callouts for Fig. 5A-B are missing. Please make sure that all Figure panels are callout out (in alphabetical order) in your revised manuscript.

We now call out Fig 5A-B. Please refer to page 12 of the revised manuscript. Changes are underlined.

9. Please state in your Materials and Methods the details of the authorities granting ethics approval - including the respective reference numbers- of the experiments involving human participants and animals, instead of only citing an older reference.

We have now specified this information in more detail. Please see sections “Human Subjects” and “Animals” within the Material and Methods section. Changes are underlined.

10. Each Dataset needs to be uploaded as an individual file with its legend in a separate tab in the same Excel file.

We have adapted the datasets 1-17 as suggested and uploaded them as individual files via the submission system. Therefore, we have also deleted the link to the datasets that were made available via FigShare.

11. Please include the title of the manuscript on the first page of your Appendix file.

We have added the title as suggested.

12. The Source Data for Fig. 3J appears to be missing. Please provide the missing data or explain/clarify.

There is no Figure 3J and we also do not call out a Fig 3J. The source data for Fig 3I has already been provided.

13. Please note that EMBO press papers are accompanied online by:
A) a short (2 sentences) summary of the findings and their significance,

We have now uploaded as “related manuscript file” a short summary file.

14. B) 2-5 short bullet points highlighting the key results

We have now uploaded as “related manuscript file” a short bullet points file.

15. a synopsis image that is exactly 550 pixels wide and 300-600 pixels high (the height is variable). You can either show a model or key data in the synopsis image. Please note that the text needs to be readable at the final size.

Please upload this information along with your revised manuscript (the text for A and B should be provided in a separate Word file).

We have now uploaded as “synopsis image”

16. - Please note that a separate "Data Information" section is required in the legends of Figures 1b-c; 2a-j; 3d-e, i; 4b-l; EV 4a-b, d; EV 5a-c. For more information and an example, please see our guide: <https://www.embopress.org/page/journal/14602075/authorguide#figureformat>.

We have now adapted all figure legends according to the guidelines provided above and the specific editorial requests.

17. Please note that the Figure panel 4g is not labelled in the Figure, however the corresponding legend for the same is labelled as 4g. This needs to be rectified.

We have added the corresponding label to the figure.

18. Please indicate the statistical test used for data analysis in the legends of Figures 2f-i; 3c-e, h; EV 2b.

We have now added this information. Please see the legends of Figures 2, 3 and Figure EV2b. Changes are underlined.

19. Please note that information related to "n" is missing in the legends of Figures 1e; 2f; EV 4a-b, d.

We have now added this information. Please see the legends of Figures 1, 2, 3 and Figure EV4. Changes are underlined.

20. Although "n" is provided, please describe the nature of the entity for "n" in the legends of figures 1b-c; EV 4a-c.

We have now added this information. Please see the legends of Figures 1 and Figure EV4. Changes are underlined.

21. Please note that the error bar is not defined in the legend of Figure EV 3c.

We have now added this information. Please see the legends of Figures EV3. Changes are underlined.

22. - Please note that the measure of center for the error bars needs to be defined in the legends of Figures 2a-f, j; 3d-e, i; 4b-l.

In all of these data the center of the error bars always represents the mean. We have now specified this as suggested. Please see the corresponding "Data information" section within the legends for Figs 2, 3 and 4.

23. - Please note that scale bar and its definition are missing for Figure EV 4c.

We have now added the scale bar to the revised Figure EV4.

24. - The main and EV Figure legends should be moved after the References.

We moved the figures' legends as requested.

Response to the reviewer comments

Reviewer #1 & Reviewer #3:

Reviewers 1 and 3 find our revised manuscript suitable for publication and further highlight the novelty and significance of our findings. They have no further comments.

Referee #2:

He/she says:

"In the rebuttal letter, the authors addressed all points previously raised in an appropriate and comprehensive manner, except for comment 1 (further in vivo evidence for miR-99 effects) and comment 7 (surgery efficiency), which were only partially dealt with. In the case of comment 1, we understand the legal and practical difficulties of having to perform a new batch of surgeries, and we appreciate the new in vitro data provided instead. For comment 7, we find the new data from hippocampi of mice that underwent surgery convincing enough for the anti-miR99 experiment. However, for the miR99 + Zbp1 ASO experiment, the authors only show Zbp1 levels in hippocampi. For this experiment, it would be important to show that the expression levels of both miR99 and Zbp1 is down in the region targeted (PFC) and unchanged in the control region (hippocampus). Furthermore, we noticed some typos in the legend for Appendix Fig S5 and in the main text (for example, authors refer to Figure 1F but it should be Figure 1G instead). Finally, we acknowledge that the quality of the figures and readability of the main text has substantially improved. Overall, we considered ourselves satisfied with the responses provided to our previous comments and with the work carried out during this revision."

We appreciate that this reviewer is now *"overall...satisfied with the responses provided to ... previous comments"* and understand there are two minor issues he/she likes us to address:

1.

"...However, for the miR99 + Zbp1 ASO experiment, the authors only show Zbp1 levels in hippocampi. For this experiment, it would be important to show that the expression levels of both miR99 and Zbp1 is down in the region targeted (PFC) and unchanged in the control region (hippocampus)..."

We understand that the reviewer would like us to confirm once more the efficacy of anti-miR-99b and Zbp1-ASO treatment. To briefly recapitulate, in Fig 2A, we demonstrate that administration of anti-miR-99b decreased miR-99b-5p levels, and we addressed the potential mode of action as requested by the reviewers within Fig EV3. Additionally, in Fig. 4B, C, we show that miR-99b-5p regulates Zbp1 and that anti-miR-99b increases the levels of Zbp1. Furthermore, we illustrate the efficacy of Zbp1 ASOs in decreasing Zbp1 levels in Appendix Fig. S5. We previously demonstrated these treatments do not affect miR-99b-5p and Zbp1 levels in the hippocampus when injections were performed in the PFC, as shown in Fig EV3.

This reviewer is now specifically asking for further confirmation of the efficacy of such treatments when anti-miR-99b and Zbp1-ASOs (anti-miR-99b/Zbp1-ASO group) were co-injected which we addressed previously in Fig EV3. We have now reorganized the presentation of the data and added some additional qPCR results to clarify this.

In brief, qPCR analysis for miR-99b-5p and *Zbp1* from PFC tissue of mice injected with either sc-control, anti-miR-99b or anti-miR-99b/*Zbp1*-ASO shows that miR-99b-5p levels decrease in the anti-miR-99b and anti-miR-99b/*Zbp1*-ASO groups in comparison to control. As expected, *Zbp1* levels increase in the anti-miR-99b when compared to the control group, while this effect is no longer observed in the anti-miR-99b/*Zbp1*-ASO group. These findings align with our previous observations and are now presented in an easier-to-understand format within Fig. EV3 panels D to E. Please see the revised Fig EV3 and the corresponding legend. We call-out these figure panels on page 12 of the revised manuscript. Changes are underlined.

2. We noticed some typos in the legend for Appendix Fig S5 and in the main text (for example, authors refer to Figure 1F but it should be Figure 1G instead)

We corrected the legend for Appendix Fig S5 and the typos in the main text.

Dear Andre,

I am pleased to inform you that your manuscript has been accepted for publication in The EMBO Journal.

Yours sincerely,
